# Advanced Electrocatalysts for the Oxygen Evolution Reaction: From Single- to Multielement Materials

América Higareda [1], Diana Laura Hernández-Arellano [2], Luis Carlos Ordoñez [1,*], Romeli Barbosa [1] and Nicolas Alonso-Vante [3,*]

[1] Unidad de Energía Renovable, Centro de Investigación Científica de Yucatán (CICY) Carretera Sierra Papacal–Chuburná Puerto, km 5., Sierra Papacal 97302, Yucatán, Mexico; alhigareda90@gmail.com (A.H.); romeli.barbosa@cicy.mx (R.B.)

[2] Unidad Morelia del Instituto de Investigaciones en Materiales, Universidad Nacional Autónoma de México, Antigua Carretera a Pátzcuaro No. 8701, Col. Ex Hacienda de San José de la Huerta, Morelia 58190, Michoacán, Mexico; diana_lha@outlook.es

[3] IC2MP, UMR CNRS 7285, University of Poitiers, 4 rue Michel Brunet, 86072 Poitiers, France

[*] Correspondence: lcol@cicy.mx (L.C.O.); nicolas.alonso.vante@univ-poitiers.fr (N.A.-V.)

**Abstract:** The proton exchange membrane water electrolyzer (PEM-WE) is a well-known green technology for hydrogen production. The main obstacle to its development, on a large scale, is the sluggish kinetics of the oxygen evolution reaction (OER). At present, the design of acid-stable electrocatalysts with low overpotential and excellent stability for the OER constitutes an important activity in electrocatalysis. This review presents an analysis of the fundamentals and strategies for the design of advanced electrocatalysts for oxygen evolution, reaction mechanisms, and OER descriptors. The scrutiny of OER electrocatalysts, with elemental composition from single- to multielemental, are presented. In addition, the purpose of high-entropy alloys (HEAs), a recent research strategy, for the design of advanced materials is summarized. Briefly, the effect of support materials, which are beneficial for modulating the electronic properties of catalysts, is presented. Finally, the prospects for the development of acidic OER electrocatalysts are given.

**Keywords:** oxygen evolution reaction (OER); acidic media; multimetallic active sites; support effect



## 1. Introduction

A hydrogen-based economy is a way to move towards a sustainable society driven by the increasing demand for energy due to population growth and economic underdevelopment, together with the depletion of fossil fuels and the consequences caused by their continuous and uncontrolled combustion. The accumulation of, mainly, carbon dioxide ($CO_2$) emissions in the atmosphere during the last decades caused an imbalance in the regulation of the planet's temperature, accelerating climate change and causing various human health problems. Counteracting the emission of harmful by-products and greenhouse gases, which pollute the environment in every possible way, requires the development and deployment of efficient and sustainable low-cost renewable energy systems to produce carbon-neutral energy on a scale proportional to or greater than the current energy supply, thus reducing day by day the rate of nonrenewable energy consumption until we eliminate our dependence on fossil energy resources.

Of the various renewable energy sources, solar energy is the cleanest and most inexhaustible that can be used; it requires 1 h and 24 min to provide to the earth all the energy consumed by humans ($6.04 \times 10^{20}$ J in 2022) in an entire year, according to the statistics of global primary energy consumption [1]. However, solar energy cannot be a primary source for society due to its intermittent nature and regional or seasonal variability. One promising scenario is the environmentally friendly production of hydrogen ($H_2$) as an energy carrier. In this regard, the most suitable and reliable approach to producing $H_2$ as a carbon-free

fuel is to use a water-splitting electrolyzer coupled to renewable energy. The production of green $H_2$ solves the problem of varying supply from renewable energy sources such as solar, thus providing energy that is available on demand. It involves storing excess electricity produced by renewable sources by rearranging the chemical bonds of water into $H_2$ and oxygen ($O_2$) by electricity-driven water splitting. $H_2$ can store a greater amount of energy per unit weight due to its high energy density, which is nearly threefold higher than the gravimetric energy density of gasoline [2]. Subsequently, $H_2$ could be recombined with $O_2$ in fuel cells, regenerating electricity. This system closes the water cycle in a carbon-neutral way. In addition, $H_2$, as an energy carrier, can also be used as a raw material in chemical industry processes. The conversion of renewable energy into power-to-X is emerging as a viable platform for storing excess renewable energy through the production of green fuels and high-value-added chemicals (e.g., hydrocarbons, alcohols, ammonia, etc.) [3] with minimal impact on the environment; e.g., ammonia is a basic component of fertilizers, resulting in a significant reduction in $CO_2$ emissions.

The energy system involving the production of $H_2$ leads to the study of two electrochemical half-reactions, the hydrogen evolution reaction (HER) and the oxygen evolution reaction (OER), that each take place at an electrode of an electrolyzer with an electrolytic medium for ionic transport from one electrode to another electrode. The most common types of electrolyzers, depending on the nature of the electrolyte, are alkaline water electrolyzers (AWE) and proton exchange membrane water electrolyzers (PEM-WE). Their half-cell reactions are displayed in Table 1.

**Table 1.** The half-cell reactions of a water electrolyzer in acid and alkaline media.

| Electrolyte | Half-Cell Reaction | |
|:---:|:---:|:---:|
| | Hydrogen evolution reaction (HER) | |
| Acid | $2H^+ + 2e^- \rightleftarrows H_2$ | (1) |
| Alkaline | $2H_2O + 2e^- \rightleftarrows H_2 + 2OH^-$ | (2) |
| | Oxygen evolution reaction (OER) | |
| Acid | $2H_2O \rightleftarrows O_2 + 4H^+ + 4e^-$ | (3) |
| Alkaline | $4OH^- \rightleftarrows O_2 + 2H_2O + 4e^-$ | (4) |

When comparing alkaline to acidic conditions, the PEM-WE offers advantages such as higher voltage efficiency (80–90%), high current densities (>2 A/cm$^2$), and high gas purity (99.999%) [4], in addition to a wider operating temperature, compact design, lower ohmic losses, and faster response to variable power offered by renewable energy sources [5,6]. The fabrication of a single cell is straightforward; this is favored by the proton exchange membrane (PEM) component. Not only the Nafion® 117 polymer electrolyte developed by DuPont, which is the most common, but also other types of PEMs are well-developed and readily available. In addition, a PEM ensures less gas crossover, decreasing the risk of explosion, and it also provides a higher proton conductivity (350 S cm$^2$ mol$^{-1}$) than the hydroxide ion conductivity (198 S cm$^2$ mol$^{-1}$) of alkaline solid polymer electrolytes [7,8].

After many years of research, the PEM-WE, considered the most promising water electrolysis method, is one of the mature technologies, with an estimated lifetime between 60,000 and 100,000 h [9]. Although these numbers are encouraging, these performances are often achieved under very specific conditions that require stability during application. Stability refers to the ability to maintain performance under constant current/voltage conditions and identify the cause of the initial rapid loss of performance [10]. Stability can be increased by improving the various components of the electrolyzers, such as the polymer membrane structure, and through the use of better electrocatalysts. This review focuses on the role of electrocatalysts in the anodic oxygen evolution reaction.

In general, the mode of operation of a PEM-WE, Figure 1, is to continuously feed water from the anode electrode side, which passes through the flow channels of the separator plates. Water is evenly distributed through the current collector until it reaches the anodic

catalyst layer. In the catalytic layer, OER occurs, in which water molecules ($H_2O$) are decomposed into oxygen ($O_2$), protons ($H^+$), and electrons ($e^-$). The protons cross through the PEM, and the $e^-$ are conducted through an external electric circuit. $H^+$ and $e^-$ species reach the cathodic catalytic layer to interact and form $H_2$. The use of Nafion® as a PEM provides an acidity close to 1.0 M $H_2SO_4$ (pH between 0 and 3), so the catalytic layer must be stable under acidic corrosion conditions, as cell components such as the current collector and plates [8].

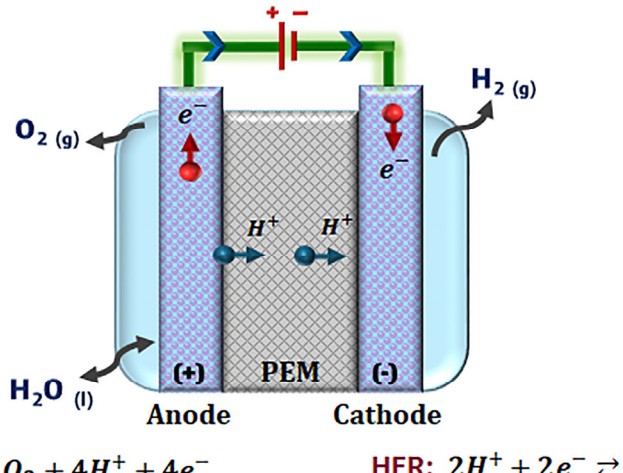

**OER:** $2H_2O \rightleftarrows O_2 + 4H^+ + 4e^-$        **HER:** $2H^+ + 2e^- \rightleftarrows H_2$

**Figure 1.** Schematic illustration of a proton exchange membrane water electrolyzer (PEM-WE).

The electrochemical water splitting requires highly efficient electrocatalysts to perform electrochemical conversions with fast but controlled kinetics, facilitating the necessary electron transfer and favoring the formation and breaking of chemical bonds. The HER at the cathode, a reduction process involving the transfer of two electrons, benefits from fast acid kinetics due to a high proton concentration [8]. The standard Nernstian potential of the HER is 0 V vs. SHE (standard hydrogen electrode); however, the applied potential is usually higher than the equilibrium potential. This difference is called overpotential (η) [11], one of the most important parameters used to evaluate the catalytic performance. The HER is best catalyzed by platinum (Pt). Despite the scarcity of Pt, low Pt loadings in the catalyst formulation (0.5–1.0 mg/cm$^2$) reveal reasonable HER activity; at applied overpotentials of about 50 mV vs. RHE (reversible hydrogen electrode), it reaches 10 mA cm$^{-2}$ [12]. The current density of 10 mA cm$^{-2}_{\text{geo}}$ (per geometric area) is a metric value corresponding to the current density expected by a solar-to-fuel device with an efficiency of about 10% under the illumination of one sun [13,14].

Water oxidation is one of the most critical electron-donating counter reactions [7]. It is a multiple sequential four-electron–proton coupled process involving the transfer of one electron at each step of the reaction, the breaking of H–O bonds, and the formation of O-O, which results in the origin of certain intermediate species with specific energy binding strength at the surface of the electrocatalyst. Kinetically, this reaction is slower by two orders of magnitude than the HER rate. This means that the energy requirement at each step results in a potential much higher than the thermodynamic redox potential (1.23 V vs. SHE) under the standard conditions (pH = 0, potential ($U$) = 0, pressure ($p_{H_2}$) = 1 bar, and temperature ($T$) = 298 K). The minimum OER overpotential for 10 mA cm$^{-2}$ is around 300 to 400 mV vs. RHE, which is still far from the requirement of an ideal electrocatalyst in terms of activity. It is reflected in that it uses more energy and demonstrates poor energy conversion efficiency. Consequently, the OER is the critical bottleneck in determining the applied voltage for overall water electrolysis.

The ideal catalyst for the OER would be a material in which each of the four steps has the same free energy change [15]. Thus, water splitting would be feasible just above the equilibrium potential [16], providing high current densities. Furthermore, such material

is based on earth-abundant elements with long-term stability in strongly oxidizing environments [17]. The corrosive conditions in acidic electrolytes make the OER an important scientific and practical challenge. So far, ruthenium (Ru) and iridium (Ir) oxides have been widely investigated as OER electrocatalysts in acidic environments due to their inherent advantageous electronic properties. These materials possess high activity and moderate resistance to acid corrosion [18]. $IrO_2$ exhibits lower catalytic activity than $RuO_2$ but higher stability [4,8]. Therefore, $IrO_2$ has been applied. However, the scarcity and high cost of these two types of material make it difficult to expand the application level of the PEM-WE [19]. In this scenario, the energy transition using green hydrogen produced by electrochemical water splitting and renewable energies is certainly attractive for a sustainable future. However, the cost of producing $H_2$ by water electrolysis is higher than producing it from fossil fuels [4], so several key hurdles must be overcome for this technology to be economically viable. In particular, the OER must achieve high current density (>500 mA cm$^{-2}$) at relatively low potentials (<300 mV vs. RHE) to offset electrocatalysts costs, and achieve long lifetimes and high energy efficiency to minimize electricity costs.

It is essential to explore new highly active, cost-effective, electrolytic-corrosion-resistant, and durable OER electrodes that can reduce OER overpotential and improve the efficiency of water splitting. The most abundant elements on earth are promising components due to their low cost and remarkable catalytic properties in strong alkaline media [17]. Transition metals with different oxidation states and coordination environments, such as tetrahedral and octahedral sites, give rise to fully tunable OER behaviors [20]. However, they are prone to corrosion in acidic media. Only a few oxides can retain their high activity at neutral or acidic pH. At low pH values, the anodic potential (1.8–2.2 V) that drives the OER usually makes these oxides vulnerable to dissolution [6], and the metal centers tend to aggregate, degrading the electrocatalytic activity and stability. A relevant factor in driving catalytic activity and stability is the composition of the material. In this sense, a new class of number-dependent materials is emerging as a research hotspot due to interesting features: high-entropy alloys (HEAs).

This review aims to provide an overview to drive the development of advanced OER electrocatalysts. Initially, the focus is on the fundamentals of OER, from the old paradigms related to the energy-scaling relation to novel strategies that aim to overcome the intrinsic limitations imposed by the conventional adsorbate evolution mechanism (AEM). We consider the fundamentals as the basis of obtaining access to different tools that allow a rational design of advanced electrocatalysts, achieving high activity together with long-term stability under acidic conditions, and, from there, exploring new perspectives of materials design. In addition, the path taken so far towards the different types of electrocatalysts that have been synthesized depending on their elemental composition is presented, i.e., from a single component to reaching the multicomponent material: HEAs. A brief section on the effect of support material on the enhancement of stability and catalytic activity is also included. Finally, opportunities and perspectives for further progress in the development of electrocatalysts for the OER are given.

## 2. Fundamentals of Advanced Electrocatalyst Design for Oxygen Evolution Reaction

The high performance of the advanced electrocatalysts depends on the number and intrinsic activity of catalytic active sites [21,22]. Strategies to increase the catalytic activities are based on the fundamentals of OER, e.g., morphology control and particle size reduction through the application of nanotechnology. Other strategies involve engineering for structure design, phase engineering, and surface reconstruction in active species. In addition, heteroatom doping or defect engineering are strategies focused on improving catalytic performance at the atomic level [23,24]. Synthesis methods play a key role in having good control and uniformity in the structure of the active phase. Numerous studies on the synthesis of nanostructured materials as heterogeneous electrocatalysts were reported [25–29]. However, despite great efforts and advances in the development of active electrocatalysts, the OER mechanism remains a research paradigm for both acidic and

alkaline environments. Although new perspectives have recently been explored, what more is needed? Paradigm shifts are required to go beyond what has already been investigated, such as exploring new approaches for the rational design of materials with superior electrocatalytic properties.

### 2.1. Mechanism of Adsorbate Evolution and Its Scaling Relationship

Based on kinetic studies, mechanistic steps have been proposed, namely, the oxide pathway, the electrochemical oxide pathway, and the electrochemical metal peroxide pathway [7,8]. The OER process was discussed in the late 1960s; however, it was later recognized by thermodynamics calculations in which it was considered a process in which $O_2$ molecules are formed by an associative mechanism rather than by the direct recombination of oxygen atoms. Density functional theory (DFT) calculations showed that there is a large activation barrier associated with direct recombination on noble-transition-metal surfaces [16].

The conventional mechanism known as the adsorbate evolution mechanism (AEM) consists of four proton–electron transfer steps occurring at a single site. In an acidic electrolytic solution, Figure 2, the electrocatalytic process begins with the partial oxidation of the $H_2O$ molecule involving electron transfer from the physisorbed water molecule to the electrode surface and its deprotonation, leading to OH* adsorbed on the active site (*) of the catalytic surface. This first step of the reaction results in the formation of the intermediate OH* adsorbed species, Equation (5). This OH* species subsequently undergoes elimination by proton–electron coupling, giving rise to the intermediate O* species, Equation (6), at the surface. In the next step, a nucleophilic attack by water on O* occurs to form intermediate OOH*, Equation (7). In the last oxidation step, electron transfer from OOH* to the electrode surface along with the release of a proton to the electrolyte is coupled with the formation of $O_2$ and released to recover the active site, Equation (8) [5,30]. Thus, in the OER process, there is the simultaneous removal of a proton and an electron coupled at each step [31], where any step could limit the overall OER yield and does not depend on the pH value.

$$* + H_2O_{(l)} \rightleftarrows OH^* + H^+ + e^- \qquad \Delta G_1 \qquad (5)$$

$$OH^* \rightleftarrows O^* + H^+ + e^- \qquad\qquad \Delta G_2 \qquad (6)$$

$$O^* + H_2O \rightleftarrows OOH^* + H^+ + e^- \qquad \Delta G_3 \qquad (7)$$

$$OOH^* \rightleftarrows * + O_{2(g)} + H^+ + e^- \qquad \Delta G_4 \qquad (8)$$

where * indicates the catalytically active site.

In the AEM, oxygen-related intermediates undergo catalytic reactions at the active sites of the metal cations of the catalyst. Therefore, the AEM can be regarded as the redox reaction of metal cations [31]. Considering $IrO_2$ as the benchmark catalyst, the OER process was characterized in situ by synchrotron-radiation-based near-ambient-pressure X-ray photoelectron spectroscopy (NAP-XPS) [32]. It was found that under OER conditions, iridium oxide nanoparticles present oxide and hydroxide species coexisting on the surface simultaneously. There is a potential-dependent change in the oxidation state of iridium from $Ir^{4+}$ to $Ir^{5+}$ predominantly on the catalyst surface, showing that $Ir^{5+}$ plays an essential role in OER, and this valence state change was related to the reduction in the number of hydroxides present on the catalyst surface [32]. These results showed experimental evidence for the DFT-predicted AEM that $IrO_2$-catalyzed OER involved an OOH-mediated deprotonation mechanism.

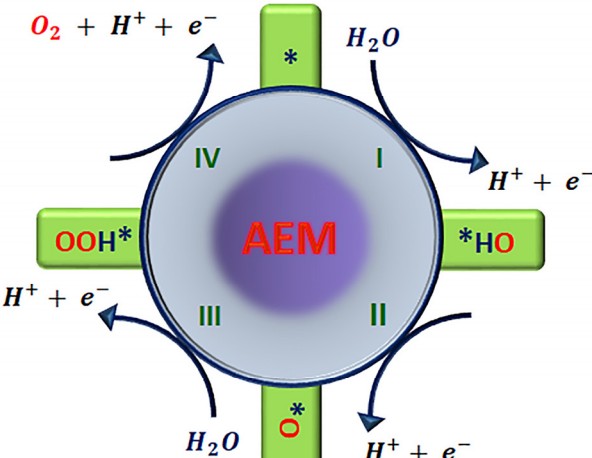

**Figure 2.** Proposed adsorbate evolution mechanism (AEM) for the oxygen evolution reaction in acid media. * indicates the catalytically active site.

According to the AEM, OER involves four consecutive electron transfer steps, and this leaves three main adsorbed oxygen-related intermediates species (OH*, O*, and OOH*), and the formation of $O_2$ requires stabilization of such intermediate species in the active site. Therefore, experimental analysis alone cannot evaluate each step of the reaction [33]. The total Gibbs free energy ($\Delta G$) of the OER was determined to be 4.92 eV ($4 \times 1.23$ eV). Although the sum of the Gibbs free energy of each elementary reaction step ($\Delta G_1$, $\Delta G_2$, $\Delta G_3$, $\Delta G_4$) is equal to the $\Delta G$ of the whole reaction, the $\Delta G$ of each reaction step can differ greatly due to the binding energies of each intermediate, which is a function of the electrocatalyst. This means that the free energies of the intermediates, relative to each other, determine the energy lost in the energy conversion to hydrogen. The step with higher $\Delta G$, in which the most energy is consumed, will be the rate-determining potential step (DPS) that controls the overall performance of the OER [8], determining the theoretical overpotential ($\eta_{theory}$). Therefore, the $\Delta G$ of the DPS should be reduced as much as possible in the electrocatalyst design. The deprotonation of OH* to O* ($\Delta G_2$) or the formation of OOH* involving the splitting of a water molecule on an adsorbed oxygen atom O* ($\Delta G_3$), which is only possible on metal surfaces that are (partly) oxidized [16], are the DPSs in most cases. If the oxygen is too tightly bound, the formation of OOH* will limit the overall reaction, but if it is too weak, the DPS will be the deprotonation of OH* [34].

It is well-known that the adsorption-free energies (E) between OH* ($E_{OH*}$), OOH* ($E_{OOH*}$), and O* ($E_{O*}$) species are linearly correlated because each of these species adsorbs on the surface through an oxygen atom [31,35]. Physically, this translates into similar bonds formed between surface-bound OH* and surface-bound OOH* [36]. Fundamentally, the slope in the linear relation of the adsorption-free energies depends strongly on the number of valence electrons of the atoms bound to the surface [37]. $E_{OOH*}/E_{OH*}$ shows a slope of approximately 1 because OH* and OOH* need one electron to fulfill the octet rule and are single-bonded [23,35]. Similarly, both $E_{OOH*}/E_{O*}$ and $E_{OH*}/E_{O*}$ have slopes close to $\frac{1}{2}$, since O* is doubly bonded to the surface and needs two electrons [16,31]. The observed strong linear correlation between different adsorption energies, the so-called scaling relation, has the effect that the adsorption-free energies of oxygen adsorbates (OH*, O*, OOH*) cannot be adjusted independently on a catalyst composed of a single type of binding site [35,38], and this occurs universally on metal and metal oxide surfaces (including perovskite, spinel, rutile, rock salt, and bixbyite, etc.) regardless of the strength of the bond to the surface. This sets a limitation of the minimum overpotential [12] for AEM-based electrocatalysts. The $\Delta E_{OOH*} - \Delta E_{OH*}$ is set to a constant value of $3.2 \pm 0.2$ eV, Figure 3. This scaling relation implies that the $\Delta E_{OOH*}$ value can be obtained directly from the calculated $\Delta E_{OH*}$ or vice versa, thus reducing the required computational facilities to evaluate the activity of a given electrocatalyst [23]. Moreover, the difference

between $\Delta G_{OH^*}$ and $\Delta G_{OOH^*}$ is larger than the thermodynamically ideal value of 2.46 eV ($2 \times 1.23$ eV) [39]. Consequently, the minimum theoretical overpotential is estimated to be $\approx 0.37$ eV ((3.2–2.46 eV)/2), even for the most reactive catalyst.

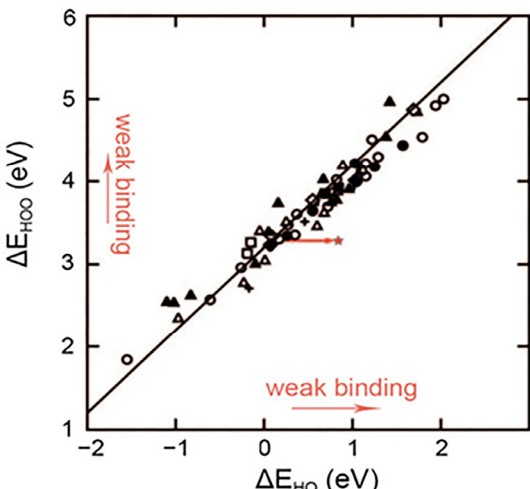

**Figure 3.** The relationship between the binding energies of the adsorbed OOH* and OH* intermediates on a series of oxide OER catalysts. Reproduced with permission of Ref. [40]. Copyright© 2023, *Advanced Functional Materials*.

Certain materials began to evolve molecular oxygen at a potential significantly above the equilibrium potential because OOH* was too weakly bound on the catalytic surface. The interest in changing the oxygen coverage lies in the fact that the other adsorbed species (O* and OH*) become more destabilized relative to OOH*, thus facilitating OER due to the decrease in potential required for the reaction step. However, depending on the scaling relationships, there is a limit to the quality that an oxygen-evolving electrocatalyst can achieve on an oxidized metal surface [16]. In this regard, we can visualize two scenarios. One is to optimize the binding energy under the scaling relations; however, even the best OER catalyst still has a theoretical minimum overpotential, setting a limit to the OER performance that theoretically cannot be circumvented anymore. The second is to break the OH vs. OOH scaling relation to enhance OER activity.

### 2.2. Design of Efficient Electrocatalysts Based on Scaling Relations

Electrocatalysis aims to lay the foundation for possible performance predictions made by understanding the intrinsic factors governing the process. However, fully theoretical or experimental approaches are impossible due to the lack of knowledge of which parameters are key to conferring the electrocatalytic properties. Therefore, correlations between physicochemical and electrochemical properties are often used to establish reactivity guides for predictions [41]. A systematic understanding of how the adsorption-free energies of these oxygen-containing intermediates affect the reaction steps is critical to rationalizing the origin of the overpotential for OER. It was found that the OER activity of the electrocatalyst is closely related to its surface electronic structure. Therefore, tuning the electronic structure could optimize the OER due to the adsorption strength of intermediates. However, it is very challenging to calculate the binding energies of adsorbates on the surface [42]. Several studies tried to clarify the structure–activity relationships of electronic-structure-related parameters. Descriptors, which are tools to identify high-performance electrocatalysts, were proposed and established to explain and screen the activity trend. Consequently, the rational design of catalysts with favorable electronic structures is supported.

Descriptors emerged to find a unifying approach to guide catalyst design and optimization. However, in earlier times, research was conducted on specific systems with little attempt to find links to other systems. The fundamental basis for the use of descriptors can be considered to have been established when Trasatti described how the

heat of oxide formation, $\Delta H_f$, was related to the volcano-shaped OER overpotential [41]. This relation was the first experimental proof of Sabatier's principle [43]. Decades later, Nørskov et al. [16] showed that the metal surface must be oxidized to produce $O_2$. They found that the difference in binding energy between the O* and OH* intermediates ($\Delta G_{O*} - \Delta G_{OH*}$) correlates with the $\eta_{theory}$, leading to a volcano-shaped relationship independent of the catalytic materials [44], Figure 4. Thus, $\Delta G_{O*} - \Delta G_{OH*}$ was introduced as a universal descriptor to predict the OER activity of various catalysts [23,43]. According to Sabatier's principle, absorbed O-containing species on the catalyst surface must not bind either too strongly or too weakly to exhibit optimal catalytic activity. Catalysts located at the top of the volcano have balanced binding energies and exhibit better OER performance. Conversely, weak oxygen binding results in deprotonation of OH* species as the DPS. These correspond to the materials located on the branch of the volcano plot on the right-hand side. On the other hand, the formation of OOH* species is the DPS for elements with a strong oxygen binding, located on the branch of the volcano plot at the left-hand side [23]. Thus, one can optimize the catalytic activity by approximating $\Delta G_{O*} - \Delta G_{OH*}$ towards the apex of the volcano plot [31]; for example, $RuO_2$, $Co_3O_4$, NiO, $IrO_2$, and $PtO_2$, Figure 4a, and $SrCoO_3$, $LaNiO_3$, $SrFeO_3$, and $SrNiO_3$, Figure 4b. However, in terms of its activity, even the best-known material operates at relatively high overpotentials. This limited efficiency has its origin in the scaling relationships between the free energies of key OER intermediaries. Therefore, even an optimal balance between adsorption energies inevitably produces a reaction overpotential [38].

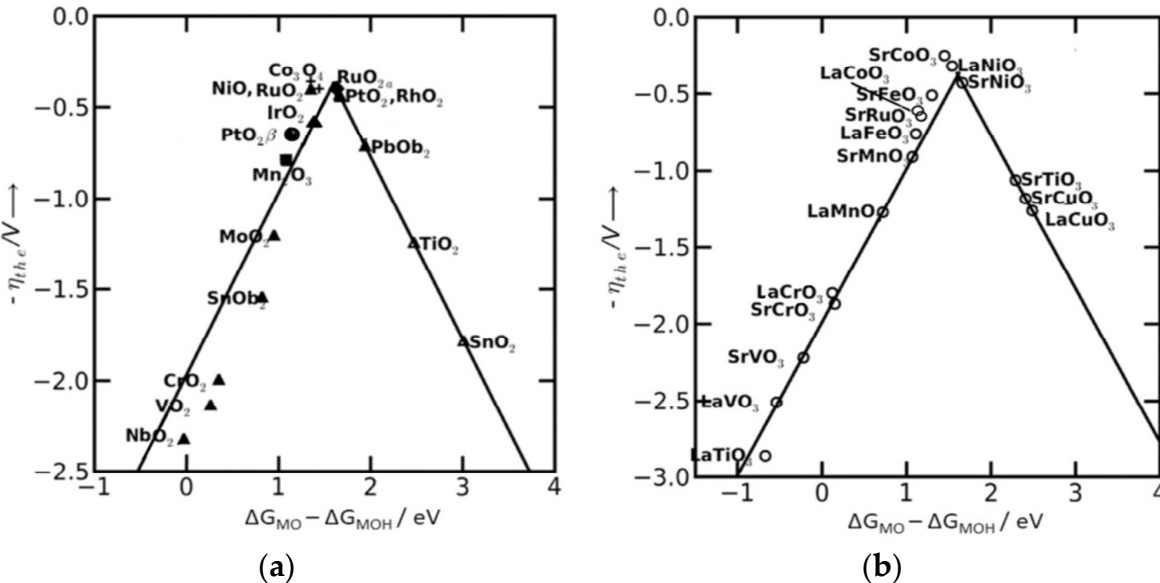

(**a**)          (**b**)

**Figure 4.** The volcano plot illustrates the relationship between the theoretical overpotential against the difference in the free energy of $\Delta G_{O*} - \Delta G_{OH*}$ on a set of OER catalysts. (**a**) Metal oxides and (**b**) perovskite oxides [45]. Reproduced with permission. Open access copyright (2023) from *Catalysis Science & Technology Journal* of the Royal Society of Chemistry.

Recent research has attempted to address the shortcomings of this approach ($\Delta G_{O*} - \Delta G_{OH*}$), as it is not well correlated for different oxidation states of the metal active site [44]. OER electrocatalysts exhibit catalytic activity strongly dependent on oxidation states, electronic properties, and local surface structure. However, due to the variable nature of different materials and dynamic surface changes, it is difficult to identify a single activity descriptor that can explain the performance of the OER. Thus, through tireless efforts over the past decades, more than 15 descriptors were found and proposed, including the *d*-band center, *p*-band center, $e_g$ occupancy, coordinatively unsaturated (CUS) metal cation, metal–oxygen bond covalency, magnetic moment, number of outer electrons, charge-transfer energy, etc., which were addressed in several reviews [23,31,37,44,46].

Although the descriptor-based approach has come a long way, each descriptor still has its limitations. Understanding the correlation between the different current descriptors of underlying physics and chemistry of OER helps to understand the mechanism better and to develop new material engineering strategies to improve the activity of electrocatalysts. In addition, machine learning and high-throughput simulations are crucial in helping discover new multiple descriptors and novel reaction mechanisms [37]. All the above descriptors are very useful for the rational design of electrocatalysts, which replaces the traditional trial-and-error approach. However, they depend on scaling relations. Therefore, efforts to find new and advanced electrode materials should be directed toward electrocatalysts that do not obey these energy-scaling relationships.

### 2.3. Strategies for Breaking Scaling Relationships

The AEM indicates a limit of activity of the OER, where the theoretical overpotential cannot be less than 0.37 eV, which is far from an ideal electrocatalyst. However, research shows that the limitations of the adsorption-energy scaling relationship in the OER, based on a conventional descriptor, can be overcome. The idea is to incorporate one or more specific active sites by detecting their functionality within the system through some descriptor or parameter that can be independently optimized. Two main pathways were studied depending on whether the OOH* is present or not. If OOH* is present, its selective stabilization is necessary without affecting the OH*adsorption. Thus, OOH* and OH* will interact differently with the catalyst surface. On the other hand, avoiding the generation of OOH* requires the activation of the electrocatalyst for a direct O-O coupling.

#### 2.3.1. OOH* Stabilization Independent on OH*

Introduction of a Second Adsorption-Relation Site

Two-site mechanisms involve reaction steps that are not considered viable reaction pathways for a single active site [43]. Having two different sites to control adsorbed OH* and OOH* alters the scaling relationship between them; this may modify the conventional AEM pathway, thus reducing the overpotential. The OER can proceed through a dual-site mechanism, with an active center for each intermediate species, leading to the adsorption-energy difference between OOH* and OH* not exceeding 3.2 eV [23], resulting in higher catalytic performance than the AEM mechanism at a single-metal site.

Formation of Hydrogen Bond (Or Introduction of Nanoscopic Confinement)

The enhancement of oxygen electrochemistry by nanoscopic confinement was recently highlighted as a promising strategy [47]. Doyle and Vojvodic et al. [36] proposed a three-dimensional catalyst structure with nanoscopic channels, which provides a confined reaction environment to allow selective interaction with the specific intermediate and favors its stabilization. This means that the introduction of a second surface to form a channel structure allows the stabilization of the intermediate species OOH* while OH* remains unchanged. The structure OOH* is more sensitive than OH*; the O in OOH* can produce an additional interaction with a different catalyst structure compared to OH*. Due to the confinement, the different interactions between intermediates and catalysts are exploited. There is a stabilizing interaction between HOO* and the O atom on the opposite surface of the channel through the formation of specific hydrogen bonds [23]. As a result, the overpotential decreases to a value below the optimal theoretical overpotential.

Introducing a Proton Acceptor (OO* + H*)

The introduction of a proton acceptor is one of the most effective approaches to stabilizing oxygen-related intermediates [48,49] in order to increase the activity of the OER. The proton-acceptor site is independent [30], so the scaling relations can be circumvented because it significantly accelerates the formation of O-O bonds [48]. Proton-acceptor sites can be formed by substituting elements with higher electronegativity that can activate oxygen on the surface. In this case, the H atom in OH* and OOH* is transferred to the

neighboring active oxygen, resulting in the formation of O* + H* and OO* + H*, respectively, which is different from the traditional scaling relation ($\Delta G_{OOH*} - \Delta G_{OH*}$ = 3.2 ± 0.2 eV); the adsorption-energy difference between OO* + H* and O* + H* is less than 3.2 eV. This results in an overpotential that is significantly lower than the theoretical minimum value. On the other hand, doping is the simplest approach, so functionalizing the catalyst surface with strong nucleophilic groups to accept hydrogen, e.g., phosphate ion (Pi) groups, can improve the interfacial proton transfer, showing significantly enhanced OER activity [23].

### 2.3.2. Direct O-O Coupling in the Absence of *OOH
Oxo Radical Coupling

The oxide path mechanism (OPM) was proposed for heterogeneous catalysts. This mechanism allows direct O-O radical coupling [50,51]. Conveniently located active metal sites work synergically to dissociate water and trigger O* radical coupling to produce O$_2$, which can be achieved without the generation of oxygen-vacancy defects or additional reaction intermediates. The only intermediates in this mechanism are O* and OH* [50].

The radical O-O coupling pathway [52] is the interaction of two neighboring metal centers at a suitable distance to allow OH* deprotonation and produce two metal–oxo species (M=O), which is the essential step for oxygen generation [53]. Next, these two metal–oxo species are coupled in a bimolecular reaction, giving rise to a peroxo species, followed by the release of O$_2$ and the turnover of the two catalytic metal centers [17], Figure 5. In principle, the O-O coupling mechanism should allow a catalyst to operate at a lower overpotential due to the absence of a scaling relation.

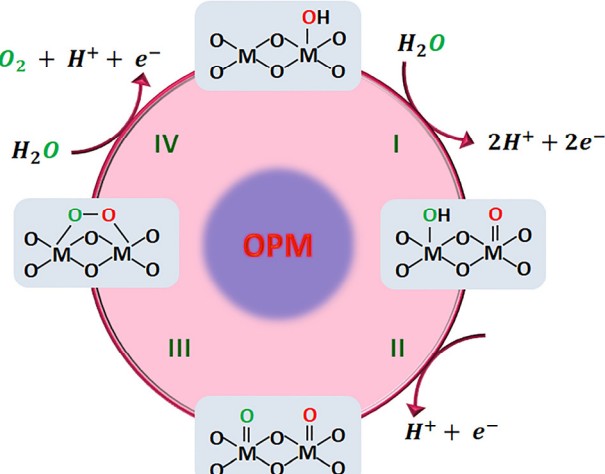

**Figure 5.** Proposed oxide path mechanism (OPM) for the oxygen evolution reaction in acid media.

Some recent investigations described electrocatalysts with activity above the theoretical limit, suggesting the presence of other reaction mechanisms. For example, arrays of Ru atoms supported by crystalline α-MnO$_2$ nanofibers were shown to be a very active and stable OER electrocatalyst in acid. Extensive research along with theoretical calculations demonstrated that OER occurs via OPM at a symmetric Ru − Ru dual site, with the key step being the direct O-O radical coupling. This reaction pathway allows Ru/MnO$_2$ to overcome the overpotential limitations of the conventional AEM, showing an ultralow overpotential of 161 mV at 10 mA cm$^{-2}$ and excellent stability for 200 h operation [50].

The construction of the interface between two or more components in heterogenous structures is a novel strategy to obtain adequate geometries and electronic structures to activate the OPM. Based on this, the Co$_3$O$_4$/CoRuO$_x$ electrocatalyst with a unique heterojunction structure, involving a spinel-type Co$_3$O$_4$ and a rutile-type CoRuO$_x$, showed an overpotential of 190 mV at a current density of 10 mA cm$^{-2}$ during an acidic OER. In operando and ex situ characterizations along with theoretical analysis supported that

Co atoms embedded in the rutile $RuO_2$ structure form Ru-O-Co sites, preventing Ru leaching during the OER process via an asymmetric dual-active site undergoing an OPM pathway [51].

The recent paradigm for O-O bond formation represents a new design principle for next-generation high-performance electrocatalysts.

Lattice-Oxygen-Mediated Mechanism

The AEM and the lattice oxygen mechanism (LOM) are different from each other in their pathways for the generation of oxygen molecules to the $O_2$ product. The AEM postulates that they are derived from water molecules. The LOM, on the other hand, proposes that the oxygen molecules are derived from the lattice oxygen of the catalyst oxides and that the key step is direct O-O coupling. The discovery of the LOM challenges the traditional view that electrocatalysis is a surface reaction [23]. Furthermore, the LOM involves nonconcerted proton–electron transfer steps that exhibit pH-dependent OER activity [8]. Although the lattice oxygen exchange reaction was reported many decades ago, the first studies involving lattice oxygen date back to 1976. Between 2007 and 2013, this mechanism was again considered as a possibility [54]. An increasing amount of experimental evidence and theoretical calculations support this reaction pathway for the OER. Here, it should be recalled that the fact that the OER entropy turnover in cryoelectrochemistry experiments was very important in $RuO_2$ with respect to Pt [55]. It is therefore a factor strongly indicative of the presence of the LOM phenomenon. Using DFT calculations, the electronic origin and feasibility of the involvement of surface lattice oxygen ($O_{surf}$) during the OER on perovskites was investigated. The involvement of $O_{surf}$ is through the nonelectrochemical pathway where adsorbed atomic oxygen (O*) diffuses from the metal site to the oxygen site, and then $O_{surf}$ moves off the surface to react with O* to form $O_{surf}$–O*, forming a surface oxygen vacancy ($V_0$) [56].

In acid electrolytes, the first two steps of the LOM are similar to the AEM, Equations (9) and (10). However, in the next step, the adsorbed O* species couples with the lattice oxygen ($O_L$) to release one $O_2$ molecule, Equation (11). This leaves a surface vacancy of oxygen ($V_O$) in the structure that must be filled. Then, dissociation of water generates OH* species that are absorbed, Equation (12), to fill $V_0$. In the final step, the adsorbed H* is removed to leave a clean active site, Equation (13), and the OER cycle is completed. Because OOH* is not an intermediate species in the LOM, water oxidation is free of scale-relation constraints. Notably, the LOM, Figure 6, features five elementary steps: four concerted proton–electron transfer (CPET) electrochemical steps, Equations (9)–(13), and one chemical step, Equation (11), which account for the pH-dependent OER activity. In contrast, the AEM of the OER involves four CPET steps on metal ions.

$$* + H_2O_{(l)} \rightarrow OH^* + H^+ + e^- \tag{9}$$

$$OH^* \rightarrow O^* + H^+ + e^- \tag{10}$$

$$O^* + O_L \rightarrow O_{2(g)} + V_O \tag{11}$$

$$V_O + * + H_2O_{(l)} \rightarrow H^* + H^+ + e^- \tag{12}$$

$$H^* \rightarrow * + H^+ + e^- \tag{13}$$

where (*) indicates the catalytically active site, which can be represented as $O - * - O$ to show the participation of the oxygen lattice, and $V_O$ is an oxygen vacancy.

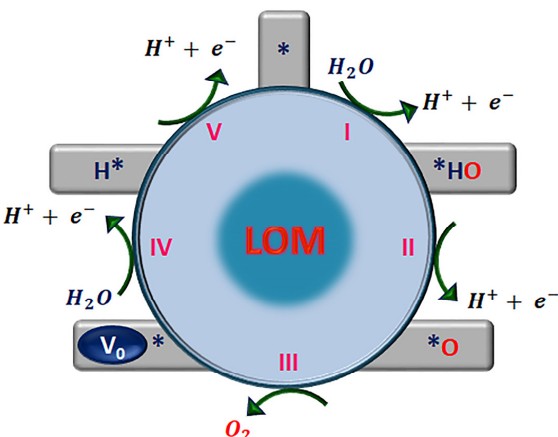

**Figure 6.** Proposed lattice oxygen mechanism (LOM) for the oxygen evolution reaction in acid media. * indicates the catalytically active site.

The LOM is considered to be superior to the AEM in increasing the OER yield because the LOM can decrease the overpotential compared with the $\eta_{theory}$ minimum defined in the AEM by having a lower reaction energy barrier. Due to favorable interaction with OH* to form oxygen vacancies and direct O-O coupling, the LOM can circumvent the scaling-relation limitation [8]. In addition, it was suggested that surface defects such as oxygen vacancies can increase the electronic conductivity of the material, improving the rate of charge transfer during the electrocatalytic process. Oxygen vacancies play an important role in the catalytic activity of the OER. The incorporation of Mo as a cocatalyst could favor the formation of oxygen vacancies due to the various oxidation states present in Mo bronzes ($H_nMoO_{3-x}$) and the mobility of protons on its surface [57]. In addition, cation vacancies can induce lattice distortion, which modifies the local electron distribution that influences the adsorption behavior of intermediate species. The formation of cation vacancies increases the number of electrophilic $O^{I-}$ species vulnerable to nucleophilic attack by water molecules or hydroxyl species with reduced kinetic barriers that contribute to the enhanced activity of the OER. Thus, oxygen vacancies facilitate water dissociation kinetics and modify the electronic structure, optimizing catalytic performance [58].

However, the LOM can lead to increased dissolution of metal active sites and thus limit the stability of the catalyst, since the reversible step towards the formation of oxygen vacancies on the surface, which plays a crucial role in the LOM, can lead to significant insertion and removal of oxygen into and out of its lattice, accompanied by the dissolution of metal cations and partial redeposition in the lattice. In fact, it was proposed that the dissolution process is triggered by the LOM. The interaction of $O_L$ and O atoms cause the structural change in the catalyst surface due to structural rearrangement [33]. Further theoretical and experimental studies and more precise in situ and/or in operando characterization techniques are still required to understand the mechanism of the LOM of the OER in-depth. The degradation processes of the catalyst active sites need to be explored. In the meantime, the best mechanism is to be determined. The AEM is limited by the binding strength of the oxygenated intermediates, which favors the study of the LOM; however, the resulting structural instability for metal dissolution and $V_0$ formation may be detrimental to long-term durability. In contrast to the AEM and LOM, the OPM is more ideal as it can break the scaling relationship without sacrificing stability, but the OPM has more stringent requirements for the geometric configuration and electronic structure of the active sites [50,51].

The following conclusions can be drawn from the fundamentals of the AEM and LOM [7,31,34]: the AEM involves four coupled proton–electron transports, so the OER activity is pH-independent. However, the LOM has a nonlinear proton–electron transfer step for which OER activity is pH-dependent. In the AEM, the observed correlation between the different adsorption energies of reaction-intermediate species and catalyst activity,

called scaling relations, leads to a volcano plot, which sets a constraint on the minimum overpotential. In contrast, the formation of OOH* intermediate species does not intervene in the LOM, which can break the scaling relations. In the AEM, all intermediate species in the reaction are adsorbed on the surface of the metal ion center, which is considered the catalytic active site; in the LOM, lattice oxygen is involved, so lattice oxygen is assumed to be the active site, and the formation of oxygen vacancies is crucial for activity enhancement.

Typically, there is not only a single mechanism involved in OER, and both the AEM and LOM have been widely accepted OER mechanisms [59]. Whether the catalytic process undergoes an AEM or a LOM, it is essential to optimize the adsorption energy of key intermediates on the catalyst by modifying the electronic structure of the catalysts based on the activity descriptors, with the goal of further enhancing the catalytic activity [33]. It should be briefly mentioned that the emergence of the LOM is concurrent with the conventional AEM based on thermodynamic and kinetic considerations. Kolpak et al. [60] also showed that both mechanisms are possible for oxides with moderate metal–oxygen covalency, but the LOM is favored, as supported by DFT calculations. The intermediates containing oxygen adsorbed on the oxide surface could originate from the electrolyte or from lattice oxygen ions, which cannot be easily distinguished during the OER process. This is because after the release of the oxygen vacancy, the lattice oxygen can be filled by the dissociation water or by the migration of bulk lattice oxygen [8]. Thus, the AEM and LOM can proceed simultaneously; therefore, there is competition between the reaction steps of each OER pathway, Figure 7. The dissolution of the metal active site and the formation of oxygen are competitive reactions; thus, the formation of oxygen molecules does not necessarily lead only to the dissolution of metal sites [61] but also to the OER. Thus, an innovative vision could take advantage of the merits of each mechanism, and both could coexist in an OER cycle to find a balance between lower overpotential (higher activity) without sacrificing long-term stability, that is, stabilizing the respective pathway intermediates to increase the selectivity towards the desired product. However, understanding the question of which physicochemical properties of catalysts determine the competition between the AEM and LOM remains elusive, which becomes a major challenge in the engineering of advanced electrocatalysts.

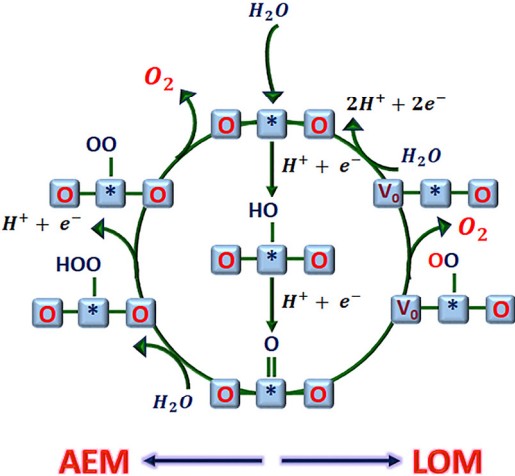

**Figure 7.** Proposed mechanism for OER in acid involves the adsorbate evolution mechanism (AEM) and lattice oxygen mechanism (LOM). $O - * - O$ indicates the catalytically active site, and $V_0$ is an oxygen vacancy.

Krtil et al. [62] found that the mechanism of LOM depends on the applied voltages. The potentials must be higher than 1.42 vs. RHE for oxygen exchange to occur at $Ru_{0.9}Ni_{0.1}O_{2-\sigma}$. Mefford et al. [34] further proposed the relationships between oxygen vacancies, M–O bond covalency, and lattice oxygen reactivity by introducing $Sr^{2+}$ into $LaCoO_3$. The metal–oxygen covalency, i.e., the strength of the M–O bond, is determined by the overlap

between the metal $d$-band and the oxygen $p$-band, which provides a basis for tuning the electronic structure and controlling the OER mechanism. When the metal $d$-band is above the oxygen $p$-band, the metal center of the oxides acts as an adsorption site and redox center [24], allowing the oxidation of water to follow the AEM. For an OER at 1.23 V vs. SHE, the redox center needs to donate or receive electrons whose energy is close to the thermodynamic potential of oxygen in water. On the other hand, when the $d$-band energy of the occupied metal is lower than the $p$-band energy of oxygen, the $p$-band electrons are transferred to the $d$-band, generating ligand holes and releasing oxygen vacancies [24]. This promotes the formation of oxygen ion species, $O_2^-$ and $O_2^{2-}$, by structural arrangement to reduce their energy and reach a steady state. Therefore, as covalence increases, the ability of metal cations to bind oxygen weakens, and direct O-O binding with reversible formation of oxygen vacancies may become favorable [23]. Thus, the OER mechanism can switch from the AEM to the LOM, and oxygen sites can be activated as the metal–oxygen covalency increases [35].

In addition to highly covalent perovskites, other materials evolved $O_2$ that may come not only from water but also from the oxide lattice, which was confirmed by mass spectrometry [23] in materials such as $RuO_2$, $IrO_2$, $Co_3O_4$, $NiCo_2O_4$, Co-Pi, Ni, and NiCo double-layered hydroxides (LDH). However, no such reaction with lattice oxygen exchange occurred on Pt in either acidic or alkaline media. The discrepancy or variability in material type indicates that lattice oxygen exchange appears to be structure- and crystallinity-dependent as well as sensitive to material composition [23]. Thus, more experimental evidence is needed to support the proposed reaction steps for the LOM. In the future, more LOM-based catalysts can be expected to increase the performance of the OER.

However, breaking energy-scaling relations sometimes cannot guarantee the improvement of electrocatalysts' performance. It is a requirement, but it is an insufficient strategy to optimize electrocatalysts due to possible inadequate adsorption energies of intermediates [23]. In this regard, other descriptors, such as the electrochemical step symmetry index (ESSI) proposed by Govindarajan et al. [37,63], emerged to ensure efficient catalytic performance after circumventing linear relationships.

### 3. Oxygen Evolution Reaction Electrocatalysts

Although HER electrocatalysts are relatively mature, materials for the OER still need to progress considerably to achieve the economic feasibility of water splitting. From the practical application point of view, OER in acidic media is more worthy of study than in alkaline media due to the rapid development of the PEM-WE and its fast response combined with renewable energy, among other major advantages. However, the main drawback of PEM electrolyzers is that, most of the time, electrocatalysts based on active compounds are not stable in acidic media, so nowadays, there is a limited range of electrocatalysts for the anode.

One of the most important considerations for OER catalysts with high activity is their long-term stability to operate at high current density in commercial applications [64].

Of the various materials that can catalyze the OER, iridium and ruthenium oxides have shown high activity and relatively good stability in acidic and alkaline environments, being particularly unique as OER catalysts under acidic conditions [32]. Although $IrO_2$ is considered the best catalyst in a local acidic environment, showing a relatively lower onset potential and a lower overpotential at 10 mA cm$^{-2}$ with respect to non-precious-metal-based electrocatalysts, Figure 8, the fact remains that the stability of these Ir-based catalysts under real operating conditions is far from satisfactory, mainly due to the possible formation of the water-soluble $IrO_4^{2-}$ anion [59,61]. These materials have some drawbacks, such as their high cost and low abundance [65], which create great difficulties for their wide commercial applications.

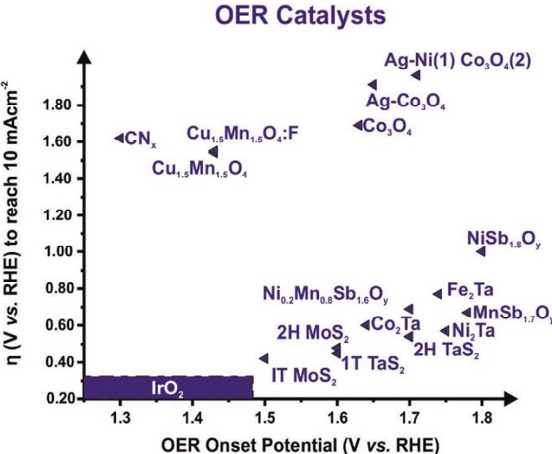

**Figure 8.** Graphical overview of OER onset potential (V vs. RHE) against the overpotential (V vs. RHE) at 10 mA cm$^{-2}$ for the non-precious-metal-based electrocatalysts and IrO$_2$ for comparison. Reproduced with permission of ref. [66]. Copyright$^{©}$ (2023), *Renewable and Sustainable Energy Reviews*, Elsevier.

Much attention has been paid to the combination of noble metals such as iridium and ruthenium with elements that are abundant in the earth. Ru- and Ir-based electrocatalysts usually show much better catalytic activity in acidic media, balancing activity and stability [33]. On the other hand, their stability can be improved by altering their surface structure [67]. However, the amount of noble metal should be reduced for a commercial application level of the PEM-WE. Consequently, these restrictions and disadvantages have stimulated the search for new, more stable, and economically viable alternatives using transition metals. In addition to their electrochemical capabilities, their structural diversity and their ability to be mixed, doped, and combined with other elements (metals and no metals) make them attractive and effective electrocatalysts [68]. Different types of transition-metal-based materials have been studied, such as phosphides, oxides, oxyhydrates, perovskites, nitrides, and sulfides. However, the strong corrosion in the acidic PEM-WE medium makes most electrocatalysts based on nonprecious metals (Ni, Co, Fe, Mn, Mo, and others) unstable. OER electrocatalysts in acidic media carry more stringent requirements than in alkaline conditions. Several studies focused mainly on finding new routes to increase the activity and stability of the catalysts specifically under acidic conditions [69].

Due to the nature of the reaction, which involves solvent decomposition causing strong acidification at the interface, and with an operating potential above 1.23 V vs. SHE, the active phase of the electrocatalyst will oxidize with time. Therefore, the materials used are mainly metal oxides/hydroxides, which are well accepted as the dominant class of chemical compounds that can resist strong-acid-corrosive conditions [31,70].

Composition optimization, good control of morphology, and nanometer particle size, among other design strategies, can modulate and accelerate catalytic activity. Substitution, incorporation, or doping with other elements in the lattice can modulate the binding energy of OER intermediates; however, the introduction of a second metal cation can result in unwanted phase separation, limiting the space for optimizing OER materials. Another strategy is to generate vacancies, which decreases the number of coordination active sites, adjusting the surface charge distribution and favoring the adsorption of reagents. In addition, adjusting the lattice strain is a useful means to alter the binding energy of the metal–oxygen bond due to changes in orbital overlap, and the designed interface not only improves charge transfer but also enhances the activity of the active site through hybridization of the active phase with the conducting support materials [23]. However, earth-abundant electrocatalysts with high activity and stability for acidic environments are still unknown [7]. In contrast, there is a wide range of transition-metal-based electrocatalysts that are applicable in alkaline environments. To develop affordable and large-scale

PEM-WE electrolyzers, it is necessary to design new OER electrocatalysts with low-cost metals of high activity and stability to replace $IrO_2$ or $RuO_2$.

To demonstrate the full performance of an electrocatalyst, it is necessary to record several electrochemical parameters. The overpotential, the potential difference at specific current density (10 mA cm$^{-2}$) so that a low overpotential indicates high catalytic activity, and the Tafel slope, which indicates how fast the current increases versus the overpotential, are the main thermodynamic and kinetic parameters, respectively. In addition, it is necessary to include activity/mass ratios, electrochemically active surface area (ECSA), turnover frequency (TOF) of catalytic sites, faraday efficiency (FE), and exchange current density ($j_0$). In addition, it is essential to assess stability by chronopotentiometry (at a fixed current density) or chronoamperometry (at a constant potential) in prolonged operation (>12 h), where minimal variation in potential or current indicates increased durability [71]. Accelerated durability testing via potential cycles of cyclic voltammetry (CV) curves, usually from hundreds to thousands of cycles at a high scan rate (100 mV s$^{-1}$), is commonly used to assess stability. In this test, linear sweep voltammetry (LSV) is carried out before and after the CV cycles to determine the potential change at a specific current density (e.g., 10 mA cm$^{-2}$).

### 3.1. Single-Component and Binary-Component Electrocatalysts and Effect of Dopants

Most earth-abundant materials that are OER-active are vulnerable to corrosion due to the acidic environment, while those that are acid-stable are inactive for the OER. Transition metals that have been identified as efficient OER catalysts include divalent cations of metals such as Mn, Fe, Co, and Ni, which exhibit electrocatalytic activity in the following order, $Mn^{2+} < Fe^{2+} < Co^{2+} < Ni^{2+}$ [72]; this is due to a weakening of the metal–oxygen bond, resulting in reduced corrosion resistance [66]. Cations with high valence states are suggested as the active site for the OER, for example, $Fe^{3+}$, $Fe^{4+}$, $Co^{3+}$, $Co^{4+}$, or $Mn^{3+}$, since the high oxidation states could serve as an electron acceptor [40]. It was shown that the really active species of transition-metal-based electrocatalysts are transformed metal oxides (M–O) or (oxy) hydroxides (M(O)OH) [24], as well as metal oxides that can be dynamically reconstituted under OER conditions.

The active site of nickel (Ni) electrodes can be nickel oxyhydroxides, NiOOH, due to the strong oxidizing condition, where an oxide state is thermodynamically favored within the OER potential window. It was reported that the electrochemical oxidation of $Ni(OH)_2$ to NiOOH results in the oxidation of Ni from $Ni^{2+}$ to $Ni^{3+}$. The decrease in the OER overpotential with an increasing amount of NiOOH indicates that $Ni^{3+}$ may be the active site [73]. However, it was shown that pure Ni cannot efficiently split water [74]. Other cocatalysts, such as iron (Fe), improve the performance of nickel oxyhydroxides. In the 1980s, it was observed that the incorporation of a trace amount of Fe in nickel oxide can significantly improve the OER [75]; also, Fe impurities greatly enhance the activities of Co-based catalysts [76]. In the case of Fe, the formation of $Fe^{2+}$ (FeO or $Fe(OH)_2$) is followed by the $Fe^{3+}/Fe^{2+}$ redox transition by the structure of $[Fe_2O_3(OH)_3(OH_2)]_3$, and then the formation of $Fe^{3+}$ species such as $Fe_2O_3$, FeOOH, and $Fe_3O_4$, and then to oxo ligands with oxidation state higher than +3 [73]. It was suggested that Fe is the active site in Fe-based compounds [77] because it can reach a high-oxidation state (high-ox). Thus, there is a clear shift in focus toward high-ox iron, and a trend of $FeO_xH_y > CoO_xH_y > NiO_xH_y$ activity has been observed. Furthermore, this has led to the proposal of single- and dual-site models with high-ox metal species as the active centers [78]. For example, Zhan et al. [79] prepared a dual Fe-Co double-layer catalyst using ZIF-8@Zn5Co1-BZIF (ZIF: zeolite imidazole frameworks) as a precursor and a dual-solvent method to incorporate $Fe^{3+}$ after a pyrolysis process. The bilayer structure formed provides abundant active sites and a stable coordination environment of the Fe-Co dual sites that prevent their sintering and regulate the adsorption-free energy of the electrochemical intermediate species. The two adjacent metal atoms redistribute the electronic structure and modify the *d*-band. The authors reported improved oxygen

reduction reaction activity and durability of Zn–air batteries. Based on its characteristics, this material could be of interest for the OER.

In this regard, NiFe-based oxide/hydroxides are considered to be the most active OER catalysts, showing higher activity compared to other materials in alkaline media, as well as showing good catalytic stability for more than 3000 h [80]. NiFe-based layered double hydroxide (LDH) material showed high cell performance with a current density of 500 mA cm$^{-2}$ at 1.68 V vs. RHE and showing good stability for 200 h in 1.0 M KOH at 70 °C in a three-electrode system [81]. A similar material, the NiFe-LDH synthesized by an electrochemical deposition method, showed good performance with a lower overpotential of 270 mV at 35 mA cm$^{-2}$ and a higher current density of 125 mA cm$^{-2}$ for the OER, showing long-time stability for 700 h in 1.0 M KOH [82]. An electron-rich NiFe-LDH electrocatalyst through an interface with conducting polyaniline (PANI) showed a low overpotential of 220 mV at a current density of 10 mA cm$^{-2}$, a small Tafel slope of 44 mV dec$^{-1}$, and good stability in 1.0 M KOH. The work demonstrated that a simple interface engineering strategy can modulate the local electronic structures of active metal sites [83].

It was observed that the method used for the synthesis can contribute greatly to the activity. Pulsed-laser ablation in liquids (PLAL) has been used to synthesize NiFe-LDH with intercalated nitrate and water. An overpotential of 260 mV at 10 mA cm$^{-2}$ was obtained on flat, highly ordered pyrolytic graphite working electrodes. The results were attributed to the morphological and structural properties obtained with PLAL. This method has attracted interest as it can control size and composition over a wide range of tunable experimental parameters [84].

In addition, the role of crystallinity has been studied. NiFe-LDH with Ni/Fe ratios from 2 to 4 and different levels of crystallinity was synthetized by hydrothermal method. This experiment showed that an increase in crystallinity gives a higher overpotential, which is detrimental to the catalytic efficiency, while the Ni/Fe ratio did not present an impact on the OER performance [85]. In another contribution, amorphous/crystalline heterostructured NiFe-based catalysts with alloy–oxide interfaces in alkaline solution were investigated. The amorphous Ni-Fe oxide with a disordered structure facilitated the formation of active oxyhydroxides under OER conditions compared to a more crystalline counterpart. This is consistent with DFT calculations indicating a lower activation energy barrier for the formation of the intermediate OOH species, causing remarkable OER activity [86].

The β-Ni(OH)$_2$ crystallizes in the brucite-type layered structure. In this structure, oxygen atoms form a hexagonal close-packed structure (HCP) in which Ni atoms are located in alternating layers of octahedral sites sandwiched between the layers of oxygen atoms [87]. However, if metal (III) cations replace the divalent nickel ions, the brucite layer structure of β-Ni(OH)$_2$ is transformed into a pyroaurite-type double-layer structure consisting of brucite layer cations, causing the higher charge of the substituent ions, and disordered interlayers in between. The transformation of the brucite structure into the pryoaurite type generally improves the electrochemical properties of Ni(OH)$_2$; however, the different substituents influence the redox mechanism quite differently [87].

NiFe-based oxides/hydroxides present a stable and suitable matrix, as Fe occupies Ni lattice sites in Ni(OH)$_2$ [74], resulting in a unique 3D electronic configuration in which the metal ions present unusual oxidation states. Therefore, during the electrochemical oxidation, Ni$^{4+}$ and Fe$^{4+}$ are formed. Therefore, the β-Ni(OH)$_2$ can act as a matrix for cations with unusual valence states. The main reason for the good catalytic stability of the binary compound Fe–Ni is attributed to the interactions between Fe and Ni and their high oxidation states [88]. Thus, the NiOOH lattice and Fe are very important to the enhancement of the activity of the electrocatalyst. The introduction of Fe can effectively modulate the local O electronic configurations of the β-Ni(OH)$_2$ and transform the OER reaction pathway [89].

DFT calculations indicate that the introduction of Fe can activate the lattice oxygen and cause electron localization around the oxygen and the Fe-O bond, generating

oxygen nonbonding states (ONB) with specific local configurations. However, there is no absolute knowledge about the nature of the active phase, which is still under debate [75,90]. Only a few studies have discussed whether Fe or Ni constitute the active sites [74]. $\beta$-NiOOH is the right type of hydroxide to catalyze the OER. In this line, Fe-doped $\beta$-Ni(OH)$_2$, formed by Fe$^{3+}$ occupying the Ni lattice sites without phase change, could be an effective electrocatalyst. However, it is difficult to prepare by traditional chemical methods [74]; therefore, few experimental investigations of OER activity have been carried out. Zhu et al. [74] synthesized crystalline samples of Fe$^{3+}$-doped $\beta$-Ni(OH)$_2$ by an atomic-scale topochemical transformation route for the first time and demonstrated that Fe$^{3+}$(0.5)-doped $\beta$-Ni(OH)$_2$ shows a current density of 10 mA cm$^{-2}$ at an overpotential of 260 mV in 0.1 M KOH solution, which is a higher OER activity than commercial IrO$_2$. It was also demonstrated that Fe(0.5)-doped $\beta$-Ni(OH)$_2$ exerts higher OER activity than Fe(0.5)-doped $\alpha$-Ni(OH)$_2$ synthesized under similar conditions. Subsequently, Kou et al. [91] reported a significant breakthrough in decreasing the overpotential for OER. Fe-doped $\beta$-Ni(OH)$_2$ nanosheets supported by Ni foam reach an overpotential of 219 mV at a geometric current density of 10 mA cm$^{-2}$ in 1.0 M KOH solution. Although these NiFe-based materials showed low overpotentials and even superior stability compared to the commercial Ir [92], they were evaluated in an alkaline medium.

In addition to an external potential, temperature contributes to the formation of oxyhydroxides. The spinel oxide NiCo$_2$O$_4$ acts as the active species at room temperature (25 °C), while at 45 °C, Ni(Co) oxyhydroxides are formed and act as the active species. The surface reconstruction of the active species from spinel to oxyhydroxide was reversible, so that it can be regulated with temperature. The results indicated a higher OER yield at 45 °C than at room temperature. In addition, DFT analysis corroborated that NiOOH has a lower overpotential than NiCo$_2$O$_4$ for OER [93]. Temperature is an important factor affecting the stability under real operating conditions [94]. Other studies have also supported the formation of NiOOH as an active phase, for example, under realistic alkaline conditions (20–30 wt% KOH) [95] or in crystalline–amorphous electrocatalysts [96], suggesting that the reconstruction process is very common in OER electrocatalysts based on Ni and other transition metals.

Surface reconstruction, although difficult to identify the active sites during this process, could be advantageous in increasing catalytic performance. In addition, it can balance activity and stability. Surface reconstruction is often related to oxidation, leaching/dissolution, and redeposition of metal cations. Therefore, a thorough understanding of the reconstruction process is necessary to synthesize a precatalyst efficiently. For example, within the OER potential region, the subnanometer Co$_3$O$_4$ layer is transformed into amorphous CoO$_x$(OH)$_y$, which is composed of di-$\mu$-oxo bridged Co$^{3+/4+}$ ions. Impressively, the formed catalytically active layer reverts to the original state upon return to the non-OER potential region [97].

Furthermore, rational design of electrocatalysts is not limited to a single strategy. For example, a systematic study was conducted to determine the role of oxygen vacancies on OER performance. The NdNiO$_3$ electrocatalyst was synthesized with different concentrations of oxygen vacancies [98]. It was determined that a balance between oxygen vacancy density and Ni$^{3+}$/Ni$^{2+}$ ratio is necessary to achieve superior OER catalytic activity. A typical volcano-shaped dependence on oxygen pressure was demonstrated, whereby an appropriate oxygen vacancy density can adjust the binding energy of oxygen intermediates.

As a complement to Fe-based compounds, for example, Co-doped hematite thin films on Ti foil showed stability for 50 h in 0.5 M H$_2$SO$_4$ and an overpotential of 650 mV at a current density of 10 mA cm$^{-2}$ [99]. Maghemite ($\gamma$-Fe$_2$O$_3$) and hematite ($\alpha$-Fe$_2$O$_3$) were synthesized as mixed films on Ti foil. These compounds were found to be active and stable for the OER in 0.5 M H$_2$SO$_4$ with an overpotential of 650 mV at 10 mA cm$^{-2}$. Hematite helps to mitigate corrosion during OER as it stabilizes the $\gamma$-Fe$_2$O$_3$, and maghemite is the active phase due to Fe vacancy [2]. This difference between their different functions is related to their different crystal structures. Titania nanowires embedded in iron oxides

on Ti foam ($Fe_2O_3$-$TiO_2$ NWs/Ti) presented an overpotential of 330 mV at 1 mA cm$^{-2}$ and a Tafel slope of 111 mV dec$^{-1}$, showing a loss of activity of ~ 20% after 20 h in 0.5 M $H_2SO_4$ [100]. Fe dispersed in N-doped carbon hollow spheres was also evaluated for OER in 0.5 M $H_2SO_4$. An overpotential of 320 mV at 10 mA cm$^{-2}$ was reported, with a Tafel slope of 300 mV dec$^{-1}$, and the Fe-NC hollow spheres showed similar acidic OER activity.

Cobalt (Co) metal has multiple oxidation states. The transition from $Co^{2+}$ ($Co(OH)_2$ or CoO phases) to $Co^{3+}$ and $Co^{4+}$ benefits OER. Thus, the significant catalytic activity of Co became a promising element for OER electrocatalysts [73]. Co-based materials were investigated because of their potential catalytic performance. However, OER was evaluated mainly using alkaline conditions, and most of these materials still exhibit a relatively high overpotential with respect to noble metals. Therefore, it is necessary to regulate the electronic structure of the active site to optimize binding to oxygen-containing intermediates and other strategies to enhance the overall activity. Optimization of the Co-based materials includes elemental doping, which can include metallic or nonmetallic elements. For example, nitrogen (N) has a high affinity to coordinate with Co ions, which facilitates interfacial transfer and reduces resistance [101]. In the case of carbon-based materials, it is known that carbon and metals can produce a strong synergistic effect, giving rise to a bifunctional electrocatalyst [102]. Along these lines, dual-element doping [102,103] has been used to adjust the valence of the active metal species, provide a synergetic effect, and increase the number of active sites. On the other hand, defect engineering at the atomic level has been used to improve electrochemical performance, including both activity and durability. This strategy is advantageous for controlling the oxygen vacancy density, effectively tuning the electronic structure, and promoting fast charge transfer [104]. In addition, it regulates intrinsic and lattice effects [101]. However, the defect concentration has to be regulated, as too many defects can hinder electron transfer [24]. In another work, because boron (B) can generate defects like oxygen vacancies, B-doped CoO nanowires were synthesized via a hydrothermal route followed by facile pyrolysis under Ar flow. The B-doped CoO electrocatalyst only needed an overpotential of 280 mV to reach 10 mA cm$^{-2}$ and present a Tafel slope of 71 mV dec$^{-1}$ under basic conditions [105]. However, the catalytic activity of Co-based electrocatalysts is lower than that of noble metals under acidic conditions.

According to previous research, among transition metals of the first row, the oxides and hydroxides of Co, Fe, and Ni are the most active for water oxidation, at least under alkaline conditions [106].

Manganese (Mn) oxide was reported to be a functionally stable oxygen evolution catalyst but showed limited activity in an acidic solution. Contrary to the increase in activity (from Mn to Ni), the stability increases in the first row of transition metals from Ni to Mn in connection with an increase in metal–oxygen interaction [66]. Mn-based materials have been investigated for their properties, such as high thermodynamic stability, suitable specific surface area, and oxidation enthalpy. In addition, Mn has numerous valence states that could be favorable in electrochemical processes since they involve serial charge transfer [107]. $MnO_2$ exhibits a wide variety of polymorphs, more than 20, including $\alpha$-$MnO_2$, $\beta$-$MnO_2$, $\gamma$-$MnO_2$, $\delta$-$MnO_2$, and amorphous manganese oxides (AMO). Their crystal structure consists of octahedral units [$MnO_6$] with oxygen atoms at the top of the six corners of the octahedron and manganese atoms in the center [108]. Depending on the structure, the stacking is different, and the connectivity between crystallographic units can be through corners or edges [109]. The catalytic OER performance in alkaline media of $MnO_2$ depends on the crystallographic structure, following an order of $\alpha$-$MnO_2$ (hollandite) > AMO > $\beta$-$MnO_2$ (pyrolusite) > $\delta$-$MnO_2$ (birnessite). The superior activity of $\alpha$-$MnO_2$ is the abundance of di-$\mu$-oxo bridges and mixed valence, benefiting charge transfer during the OER [24]. For the $\alpha$-$MnO_2$, an overpotential of 490 mV at 10 mA cm$^{-2}$ indicates a faster reaction rate relative to other structures, and a Tafel slope of 77.5 mV dec$^{-1}$ was reported in alkaline conditions [109].

The $\beta$-MnO$_2$ is considered an inert structure for oxygen conversion. To improve the catalytic activity of $\beta$-MnO$_2$, Ru$^{3+}$ ions were introduced into its lattice by a hydrothermal reaction, drastically affecting its activity and pH adaptability. The main reason for the increased OER activity was found to be the interstitial doping with Ru exposure on the surface during crystal splintering and lattice expansion [107]. DFT calculations showed that the crystal splintering exposes atomically dispersed Ru-O species on the MnO$_2$ surface as OER active sites.

MnO$_2$ is transformed to Mn$^{3+}$ during the one-electron redox process; however, Mn$^{3+}$ is unstable and undergoes a disproportionation reaction in acid/neutral solution to stable Mn$^{4+}$ and Mn$^{2+}$, which have high solubility in aqueous solutions, favoring the dissolution of the metal [110]. Mn$^{3+}$ is responsible for the adsorption of the hydroxide ion; consequently, it acts as precursor to the oxygen evolution reaction. Thus, MnO$_2$ electrocatalysts exhibit pH-dependent activity mechanisms for OER [111]. Moreover, the OER activity depends mainly on the chemical composition, crystal structure, and morphology. A recent alternative to overcome the limitations in the stability of 3D-metal electrocatalysts is based on the identification of a stable potential window in which OER can occur without corrosion [112]. The $\delta$-MnO$_2$ can be used as a long-term OER electrocatalyst under highly acidic conditions when the applied potential is between 1.6 V and 1.75 V, because at 1.8 V vs. RHE, the dissolution pathway occurs through the evolution of the MnO$_4^-$. Although dissolution of Mn$^{2+}$ occurs, preferential redeposition as $\delta$-MnO$_2$ by the self-healing phenomenon of manganese oxide was also observed [113].

Amorphous ultrathin $\delta$-MnO$_2$/$\varepsilon$-Fe$_2$O$_3$ heterojunction nanosheets with oxygen vacancies showed high-output OER activity with an overpotential of 299 and 322 mV to drive current densities of 100 and 300 mA cm$^{-2}$, respectively, and a Tafel slope of 35.1 mV dec$^{-1}$ [108]. The electrochemically induced, layered $\delta$-MnO$_2$ materials appear to be exceptional and close to the performance of precious metal oxides (Ir/RuO$_x$) under acidic conditions, such as single-phase manganese tetroxide nanoplates that provide exceptional activity, reaching a low overpotential of 210 mV in acidic media [113]. Interlayer cations play an essential role in tuning the electronic structure. Ni-intercalated birnessite (Ni$^{2+}$/$\delta$-MnO$_2$) exhibited activity superior to NiOOH or pure birnessite [114]. Molecular dynamics (MD) simulation suggested that interaction with interlayer water molecules and interlayer cations in the birnessite structure results in a unique structuring of water that enhances electron transfer reactions. Similarly, Co was intercalated into the $\delta$-MnO$_2$ interlayer, reducing the overpotential for the OER to 360 mV at 10 mA cm$^{-2}$, with a Tafel slope of 46 V dec$^{-1}$ in alkaline media.

For the above materials and other electrocatalysts summarized in Table 2 corresponding to the single and binary-component section, it can be observed that mainly metals such as Ni, Co, and Fe as single components of the OER electrode are only active in alkaline media. At the same time, Mn presents limited stability in the acid medium since it is necessary to establish a suitable potential window. Although the fabrication of binary components electrocatalysts, including Ir or Ru, presents better OER performance, the amount of noble metal is still too high for their commercial scale. On the other hand, the combination of transition metals with each other or with other elements under some design strategy improves the catalytic activity, although the stability remains limited. Thus, these materials of lower elemental composition seem unable to achieve high activity together with high stability in acidic environments [100]. In addition, metal leaching of these catalysts should be evaluated, as this effect could create a rougher surface with more active species, which would reduce stability. Overall, these one- and two-component materials represent the background; in turn, this strengthened the OER theory, which contributed to the advancement of rational electrocatalyst design.

In general, non-noble metals and their alloys must be stabilized by hosts (e.g., support materials) or saturated with elements of high electronegativity, since they cannot survive directly in aggressive acidic or alkaline media [40]. Therefore, three- and four-component

electrocatalysts, which are expected to have superior catalytic properties, are discussed in the next section.

### 3.2. Ternary and Quaternary Component Electrocatalysts

The synergistic effect of a third element, which promotes reconstruction, may be a way to tune the overall activity. This can be observed with the inactive spinel $CoAl_2O_4$ in which iron was incorporated. Iron substitution allows surface reconstruction to give rise to active Co oxyhydroxide species during OER and introduces structural flexibility that promotes the formation of oxygen vacancies [115]. $CoFe_{0.25}Al_{1.75}O_4$ presents the best performance evidence, although the result showed low current density in alkaline media. Similarly, the synthesis of $RuO_2/(CoMn)_3O_4$ with low Ru mass loading (2.51 wt%) presents an overpotential of 270 mV at 10 mA cm$^{-2}$ and a Tafel slope of 77 mV dec$^{-1}$. The introduction of Mn in the spinel oxide in $Co_3O_4$ allows charge redistribution, and the Ru species in $RuO_2/(Co, Mn)_3O_4$ are electron-rich, which are energetically more favorable sites for rate-determining step (RDS) in the acidic OER process [116].

A rational approach for the design of non-noble-metal catalysts exhibiting stability and activity under acid conditions was reported [117]. Activity and stability were treated as uncoupled elements in a mixture of metal oxides. Mn was incorporated as a stabilizing structural element, and the active phase consisted of Co centers. However, the current density is not competitive due to the potential limit, since Mn is unstable at higher potential. The CoFePbOx film was obtained following the same approach, where Pb was the structural element. Under acidic conditions, $CoFePbO_x$ exhibits a Tafel slope of 70 mV dec$^{-1}$ and presented corrosion resistance for more than 50 h of continuous cycling [117]. Thus, the prospect of decoupling functionality in mixed metal catalysts to design active, acid-stable, and earth-abundant OER catalysts could be viable. An effective tool is the Pourbaix diagram to explore the thermodynamic stability of elements as a function of pH and electrochemical potential [100,106]. The stabilizing role of structural elements, in this case Mn and Pb, in a mixture of metal oxides is a promising method to improve the stability of active transition metals [66]. It has also been shown that ternary component electrocatalysts present a wide strategy space to design materials with catalytic properties superior to those of single and binary structures.

On the other hand, intermetallic compounds (IMCs) have many advantages, such as well-ordered atomic arrangement with directed covalent interactions, a tunable coordination environment, and hierarchical cluster structures combining elements with diverging chemical properties. So far, a few examples based on two elements ($Ni_2Ta$, $Al_2Pt$) have been described [118,119]. The ternary compound $Hf_2B_2Ir_5$ was evaluated as an electrocatalyst for the OER in 0.1 M $H_2SO_4$ [120]. Ir was used due to its high performance in the OER in acidic media. This ternary compound showed a structure with the *Pbam* space group. The electrochemical system with the $Hf_2B_2Ir_5$ electrode exhibited a potential of 1.64 V vs. RHE, close to $IrO_x$ (1.61 V vs. RHE), to reach 10 mA cm$^{-2}$. After catalytic evaluation, 50 potential cycles were applied in the range of 0.05 to 1.0 V vs. RHE. $Hf_2B_2Ir_5$ showed no oxidation or reduction in the potential window, and then the OER potential dropped to 1.59 V vs. RHE, suggesting that the surface undergoes changes. A self-optimized composite of $Hf_2B_2Ir_5$ and $IrO_x(OH)_y(SO_4)_z$ showed high stability for 240 h at relatively high current densities of 100 mA cm$^{-2}$. This work supports the strategy of splitting different activity and stability functions among several phases with similar chemical composition.

Recent work with a more complex and noble-metal-free structure was reported [121], with a four-element composition, hierarchical coassembly, and interacting three-phase $MoS_2$ and $Co_9S_8$ nanosheets on $Ni_3S_2$ nanorod supported on nickel foam. The 3D composite has abundant active sites capable of catalyzing the HER and OER over a full pH range. CoMoNiS-NF-31 provides an ultralow overpotential of 228 mV for OER to reach a current density of 10 mA cm$^{-2}$ in an acidic electrolyte. In addition, it exhibits a remarkable balance between activity and stability. A synergistic effect of the charge-transfer processes between the phases is obtained. The optimized sulfide phases, $MoS_2$ and $Co_9S_8$, contribute to

improved electrocatalytic activity under alkaline, acidic, and neutral conditions for water splitting. Other electrocatalysts, corresponding to the ternary and quaternary components section, are summarized in Table 2.

A graphical summary of the catalytic performance of non-noble-metal-based OER electrocatalysts in acid media from single (e.g., $Co_3O_4$), binary (e.g., $Ag$-$Co_3O_4$-1), ternary (e.g., $Ni_{0.5}Mn_{0.5}Sb_{1.7}O_y$), and quaternary (e.g., PMFCP, Mn-doped $FeP/Co_3(PO_4)_2$) compounds is shown in Figure 9. The catalytic performance, including the Tafel slope, Figure 9a, and the overpotential to reach a current density of 10 mA cm$^{-2}$, Figure 9b, is shown. It is noteworthy that almost all electrocatalysts exhibit a higher Tafel slope value than the more advanced Ir-based electrocatalysts of 40 mV dec$^{-1}$ (dashed line inset in Figure 9a). Moreover, the overpotentials of these materials are between 350 and 700 mV at 10 mA cm$^{-2}$, so the catalytic performance of non-noble-metal-based electrocatalysts has yet to be tested to reach the level of a large-scale PEM-WE.

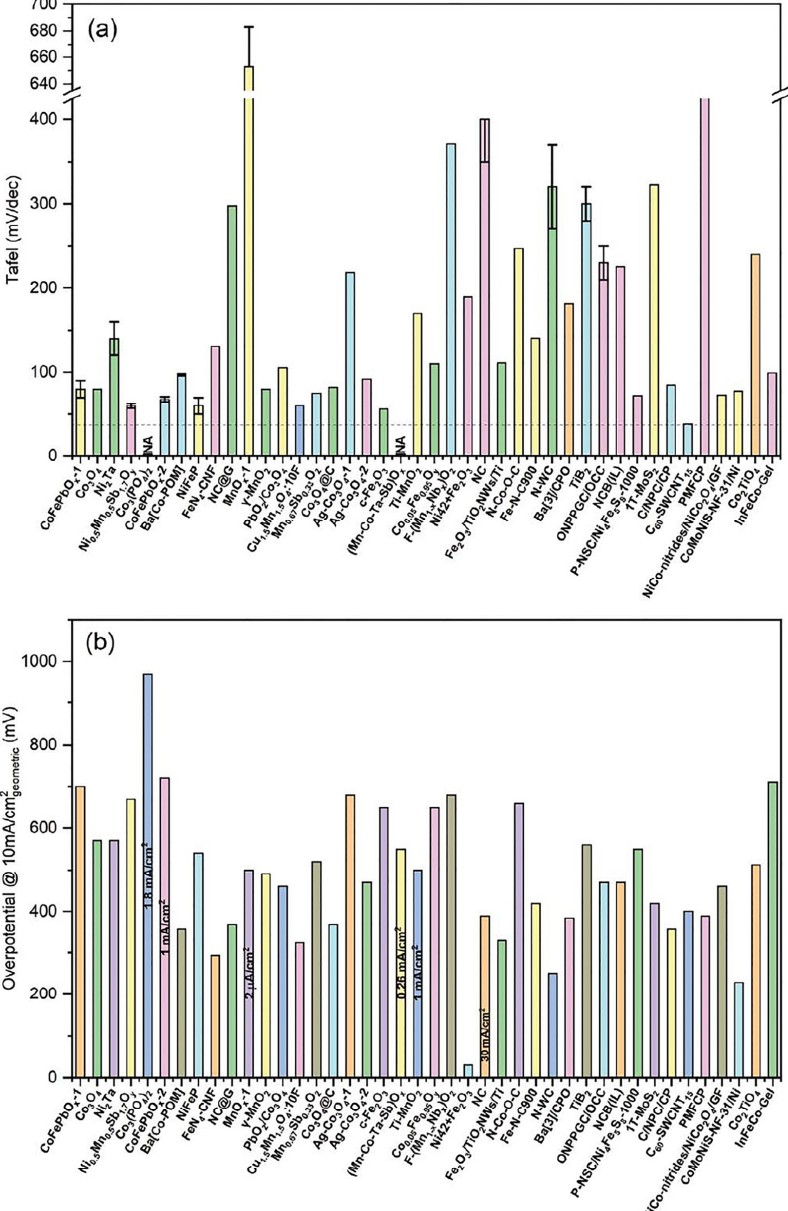

**Figure 9.** Overview of the catalytic performance of non-noble-metal-based OER catalysts in acidic media. (**a**) Tafel slope and (**b**) overpotential at 10 mA cm$^{-2}$ unless otherwise specified. Reproduced with permission from Ref. [100]. Copyright© (2023), *Advanced Materials*.

### 3.3. Multicomponent Electrocatalysts

The great efforts in the development of acid-stable electrocatalysts involving non-noble metals have led researchers to explore new approaches and changing perspectives. Recently, multielemental electrocatalysts, more specifically high-entropy alloys (HEAs), were added to the list. HEAs are promising candidates as functional materials for new applications. The nomination of an HEA is based on the high configurational entropy of the random mixture of elements that dominates the thermodynamics of solidification to form these alloys, according to the equation $\Delta S_{conf} = -R \sum_{i=1}^{n} x_i \ln(x_i)$, where $R$ is the molar constant of the gases and $x_i$ represents the mole fraction of the $i^{th}$ elemental component. As the molar ratios of the elements approach an equimolar fraction, the value of $\Delta S_{conf}$ increases, reaching a maximum. Thus, $\Delta S_{conf}$ can be simplified as $\Delta S_{conf} = R \ln(n)$. Therefore, the $\Delta S_{conf}$ calculated for equimolar alloys with 1, 2, 3, 4, and 5 elements are 0, 0.69$R$, 1.10$R$, 1.39$R$, and 1.61$R$, respectively [122,123]. The factor of 1.5$R$ is large enough to compete with the enthalpy of mixing as well as to be used as a limit in the classification of alloys, and from that value, we are in the HEA world ($\Delta S_{conf} \geq 1.5R$).

In the field of electrocatalysis, they are considered potential materials for hydrogen-evolution and oxygen-evolution reactions. In particular, their inherent properties due to elemental diversity confer them a functionality not accessible with conventional heterogeneous electrocatalysts. Thus, HEA systems are being studied as a strategy to overcome the limitations of conventional OER electrocatalysts. There is an innumerable combination when the number of components reaches five or more elements, which allows tuning the electronic properties of the electrocatalyst surface. The large elemental diversity and the incorporation of multifunctional active sites in catalysts can present near-continuous binding energy distribution patterns (BEDPs) for interaction with reaction intermediates, which are especially suitable for multistep reaction pathways such as the OER and can be a platform for breaking scaling relations. The use of a multielemental system reveals remarkable electrocatalytic performance relative to monometallic and bimetallic catalysts due to their versatile phases and excellent structural stability in harsh catalytic environments; thus, they present a valuable opportunity to reduce or even eliminate the use of precious metals while maintaining high catalytic activity [69], due to the formation of new, unique, and tailorable active sites of multiple elements in which even the coordination environment of each active atom is different; thus, strong local electronic and geometric effects are involved in the catalytic activity.

In summary, the inherent effects of HEAs, such as high entropy for thermodynamics, lattice distortion for structure, slow diffusion for kinetics, and the cocktail effect for properties, affect the microstructure, and physical and electrochemical properties of HEAs, which could be beneficial in enhancing catalytic activity, selectivity, and stability through the regulation of geometric and electronic structures. Therefore, these four core effects are useful guidelines for designing HEAs for specific purpose applications, Figure 10. It is important for the design of an alloy to understand the related factors involved before selecting suitable composition and fabrication processes.

Since HEAs are composed of several elements, there are many strategies for and approaches to tuning the catalytic properties. For example, there is the modification of the composition by incorporating several major elements (Fe, Ni, Co, Mn, Cr, etc.), the integration of carbon nanomaterials (graphene, CNT, mesoporous carbon, activated carbon, nitrogen-doped carbon, etc.) that sometimes originate during synthesis, morphology control, creation of defects such as oxygen or cation vacancies, creation of interfaces or hybrid engineering due to possible multiphases, among others, all of them coexisting and promoting a superior synergistic effect. Therefore, multimetal compounds are a promising class of OER electrocatalysts by taking advantage of the synergistic properties of metals combined with engineered structures.

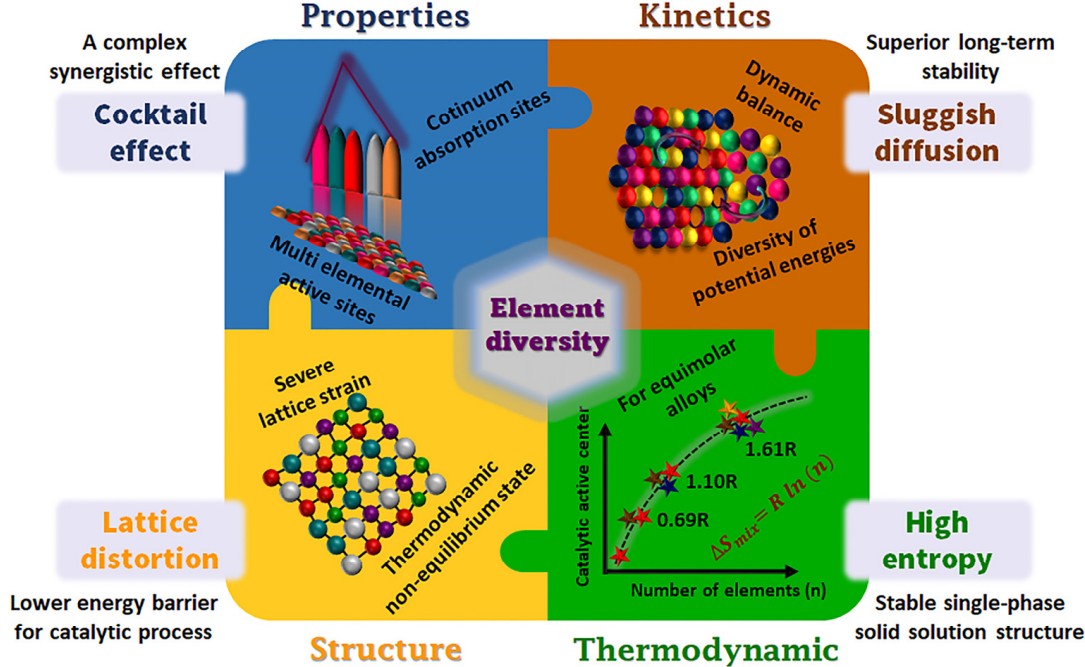

**Figure 10.** The core effects of the high-entropy alloys (HEAs). The synergistic mixture of elements results in HEAs with unexpected features as advanced electrocatalysts for OER.

The incorporation of active non-noble metals (Fe, Co, Ni, Mn, Cu, etc.) into the structure of HEAs can increase corrosion resistance and stability due to the slow diffusion of metal cations and strong lattice distortion. For example, $Ni_{20}Fe_{20}Mo_{10}Co_{35}Cr_{15}$ shows a negligible decrease in catalytic activity after long-term testing with the HER in both acidic and alkaline solutions [124]. Recently, porous CrMnFeCoNi for the hydrogenation of *p*-nitrophenol showed that all metals remained in metallic states without forming metal oxides according to the experimental results, verifying the excellent antioxidation properties of HEAs. The higher oxidation resistance, probably due to the remarkable stability of HEAs [25], may be good for use in the OER.

Specifically, for the OER, Qiao et al. [125] reported the synthesis of monometallic, trimetallic, and CoFeNiMnMoPi high-entropy phosphates (HEPi) using the high-temperature fly-through method. By controlling precursor concentration (tri-n-octylphosphine and metal precursor salts), temperature, flow rate, and other synthesis parameters, a spherical particle shape and homogeneous particle size (109 to 200 nm) were achieved. HEPi showed a lower overpotential and Tafel slope (270 mV at 10 mA cm$^{-2}$, and 75 mV dec$^{-1}$, respectively) than commercial IrO$_x$ material, with an overpotential of 340 mV at 10 mA cm$^{-2}$ and a Tafel slope of 90 mV dec$^{-1}$ for the OER in an alkaline medium. The contribution of this research is the facile, efficient, and scalable synthesis method as a new avenue for the discovery of a range of polyanionic materials.

In the synthesis of AlNiCoIrX (where X = Mo, Nb, V, Cu, Cr), the elements except Al were uniformly mixed in one single-phase nanostructure with equal atomic amounts and melted together and then remelted with a large amount of Al. The precursor alloy was chemically dealloyed in an alkaline solution to originate nanoscale pores [126]. XRD showed that the pure Al phase was dominated in the precursor alloys. The (111) peak position shifted to a higher angle; this demonstrated the successful incorporation of non-noble metals, suggesting a multicomponent alloy. The incorporation of the fifth element does not change the phase structure. STEM-EDS mapping showed good mixing of the elements. XPS showed that Ir is mainly in the metallic state due to its noble nature. For Ni, the peaks corresponded to $Ni^{2+}$. For Co 2p, the emission peaks 2p$_{3/2}$ and 2p$_{1/2}$ were assigned to $Co^{2+}$. Mo showed the highest oxidation states of $Mo^{6+}$ and $Mo^{4+}$. It was suggested that residual Al was trapped in the crystal lattice of the alloy due to a weak

signal. Oxygen can be assigned to both metal–O and metal–OH bonds. Prior to the OER test, the nanoporous HEAs were electrochemically activated by potential cycling between 0.02 and 1.2 V vs. RHE. The activation process was to remove non-noble-metal surface species, obtaining an Ir-rich surface and a mixed-oxides surface. Specifically, AlNiCoIrMo showed a low onset potential of 1.42 V vs. RHE and the lowest Tafel slope of 55.2 mV dec$^{-1}$, Figure 11. At current densities of 10 and 20 mA cm$^{-2}$, the overpotentials were 233 mV and 255 mV, respectively. The durability of these materials is also of great interest, especially in acidic solutions. Continuous CV cycling between 1.2 and 1.5 V vs. RHE was used to evaluate the durability of these samples. AlNiCoIrMo shows a slight potential increase of about 11.5 mV after 7000 cycles. After the durability test, STEM-EDS was used to evaluate the microstructure and element distribution, and this revealed that the nanoporous structure is well maintained without microstructure coarsening, and the elements did not show segregation. This excellent performance makes AlNiCoIrMo very attractive for OER in acidic media (0.5 M H$_2$SO$_4$) with the advantages of low Ir loading.

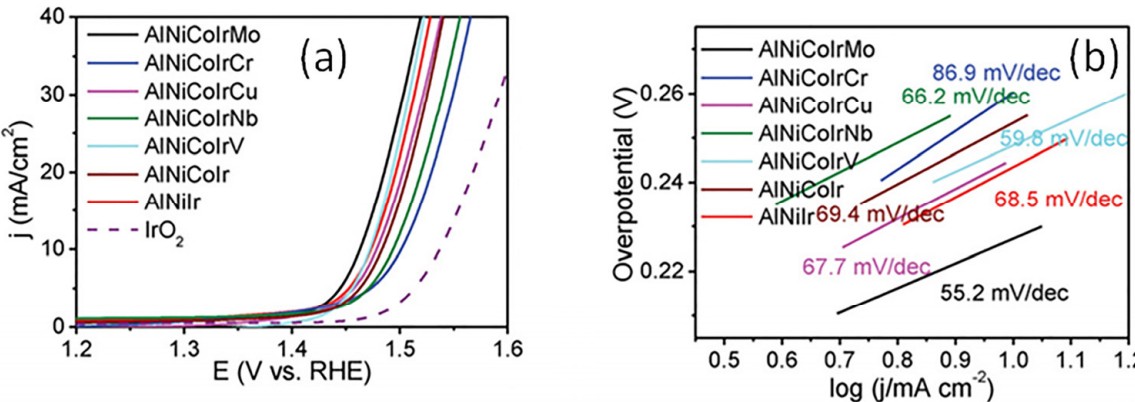

**Figure 11.** (**a**) OER polarization curves and (**b**) Tafel slope plot of AlNiCoIrX (where X = Mo, Cr, Cu, Nb, V) HEAs, with AlNiCoIr, AlNiIr, and IrO$_2$ for comparison. Reproduced with permission of Ref. [126]. Copyright$^{©}$ (2023), Small.

IrPdRhMoW/C exhibited bifunctional electrocatalytic behavior for the HER and OER. For the OER, it had an overpotential of 188 mV at 10 mA cm$^{-2}$, outperforming the commercial RuO$_2$ catalyst in 0.5 M H$_2$SO$_4$. More interestingly, in an HEA, it maintained excellent activity and structural stability for water splitting after 100 h at 100 mA cm$^{-2}$. The self-balanced effect of electronic structures in the HEA due to the coexistence of crystalline and amorphous structures maximizes the activity and stabilizes the valence states of metal sites for long durability [127].

Other electrocatalysts corresponding to the multicomponent section are summarized in Table 2. Despite all the potential that HEAs have shown, there is still much space to explore.

### 3.4. Increased Functionality of Single- to Multielement Materials

In single-element materials or bimetallic alloys, the properties are mainly determined by the principal element, and the role of dopants or cocatalysts is to adjust the binding energies. In this sense, to high-performance electrocatalysts of fewer-element composition, the base element should have very promising catalytic properties [128], implying the use of noble metals (Ir and/or Ru), and it is necessary to establish strategies to regulate the concentration of each component and doping [71]. Instead, a complex single solid solution phase also means a homogeneous distribution of all elements; therefore, multielement interactions in HEAs cause a strong divergence of the properties of the atoms as compared to a single-element surface. Thus, the reaction is no longer limited by the properties of the base element and its position in the volcano plot. Consequently, for HEAs, it is not reasonable to identify a specific ensemble as the active center as in bimetallic alloys. A

specific atom on the surface of HEAs will always be surrounded by different neighboring atoms. Therefore, there is a large amount of possible atomic arrangements on the surface of HEAs, and even the coordination environment of each active atom is different, which induces different adsorption behaviors of the active species.

The electronic structures of electrocatalysts play a key role in regulating catalytic performance, mainly including *d*-band center and charge transfer. The position of the *d*-band center of an alloy is easily regulated by varying the composition elements and corresponding concentrations, further affecting the catalytic activity and selectivity, which is also known as the ligand effect. Mainly for the surface atoms of single-metal nanocrystals, the surface charge density is essentially the same as elemental atoms on the surface. Negative electrons may accumulate slightly at the edges and corners of single-metal nanocrystals depending on morphology, size, etc. As for alloys, charge transfer between different atoms on the surface can occur due to the different work functions of the metal components, leading to significant charge redistribution on the alloy surface [25]. This charge redistribution influences the *d*-band center, helping to decrease the overpotential due to modification of the adsorption energies of key intermediates. With various composition metals, HEAs potentially possess severe redistribution of surface charges, inducing the accumulation and scarcity of electrons across the entire surface of HEAs. Thus, the charge density of each metal atom on the surface differs from that of neighboring atoms in HEAs, creating a more active center for chemical transformation. In addition, the *d*-band center modulated by charge redistribution is also able to promote catalytic performance by appropriate binding energy of the reactants. Therefore, the rational selection of the components with different work functions for the construction of HEAs is able to control the charge density on the surface of HEAs, contributing to the adsorption and activation of reactants and further improvement of the catalytic performance of HEA-based catalysts [37].

Several theorical and experimental efforts discussed that the determining feature of HEAs is the BEDPs. A distribution-of-adsorption-energy model, based on theorical calculations, demonstrated that the number of adsorption peaks equals the number of principal elements [128], considering that a specific element in the center of the active site forms one of those peaks; however, when two or three atoms are involved in the active site, the number of possible centers increases further, thus also increasing the number of adsorption peaks [129]. Similarly, by adjusting the elemental combinations and composition in HEAs, BEDPs can be tailored [130] to cover a wide range of adsorption-binding energies.

The flexibility in element selection allows for more possible combinations to match several different properties, such as optimizing two individual adsorption energies or further improving mechanical properties such as thermal stability or conductivity at the same time, resulting in multifunctional materials.

However, HEAs have some limitations relative to fewer-elemental materials, e.g., although there are a large number of diverse active sites, not all active sites contribute to functionality, so HEAs may have a lower abundance of optimal surface-active sites. In the case of HEAs, the lower probability of optimized binding energy should be compensated by higher intrinsic activity [131]. In addition, HEAs present many challenges, such as synthesis methods, which are still at a nascent stage compared to those of minor components, so efforts are needed to obtain accessible synthesis routes. Furthermore, it is essential to integrate advanced characterization techniques, comprehensive electrochemical analysis, and theorical support, as complex HEAs present enormous possible compositions for experimental analysis. Therefore, the combination of simulation techniques, machine learning, data-driven combinatorial synthesis, high throughput, and screening is highly desirable for efficient exploration of a suitable HEA composition and cost reduction [132].

On the other hand, from conventional compounds to multielements, it is worth noting that a thorough understanding of the OER mechanism, including the proposal of new routes, enables the design of robust and highly efficient OER electrocatalysts for application in functional devices. It is very important to understand the structure–activity relationship at the atomic level and under realistic working conditions using in situ techniques,

e.g., Raman spectroscopy, X-ray absorption spectroscopy (XAS), near-ambient-pressure X-ray photoelectron spectroscopy (NAP-XPS), Mössbauer spectroscopy, ultraviolet–visible (UV-vis) spectroscopy, differential electrochemical mass spectrometry (DEMS), and scanning electrochemical microscopy (SI-SECM) with surface interrogation [71,75]. Furthermore, new in situ techniques should be critically implemented to accurately determine the active site of the OER due to the multidynamic effects coexisting in the OER process, such as oxidation state, crystal structure, and morphological changes, as well as phase transformations and other disturbing factors. These variations could be the origin of the difficulty in differentiating the active center, i.e., the determination of the role of each component, since so far, no technique has the ability to capture a global view of the diversity of transformations under the OER. When such a breakthrough is achieved, it will have an impact on the design of high-performance electrocatalysts as well as on the presence of new functionalities from those of minor composition to those of complex structure.

## 4. Effect of Supports

Support materials play an essential role in improving the catalyst thermal stability, corrosion resistance, and catalytic activity by increasing their surface area, favoring interactions between the active phase and the support and reducing active surface loss such as sintering. The functionality between the support material and the nanoparticles is associated with several factors: (i) Strong metal-support interactions (SMSI) that modify the electronic properties of active sites and enhance the intrinsic activity of the catalyst [133]; they influences the surface physicochemical properties of the surface, promoting stability and oxophilic characteristics that increase tolerance to poisoning species [134]. (ii) The size conservation and distribution of the electroactive nanoparticles on the surface of the support have a strong impact on the number of their available active sites and therefore on the overall catalytic activity [135]. (iii) The support material must have a large surface area to achieve a uniform dispersion of the NPs, avoiding possible agglomeration, and have a good electronic conductivity of the electrode [136]. Therefore, the proper choice of support material is crucial and highly influential in determining the OER behavior. Currently, various organic and inorganic materials are used in the synthesis of NPs, including those based on noble metals [137–139], oxides [140–142], hydroxides [143], phosphides [144], nitrides [145], chalcogenides [146,147], alloys [148], and composites [149]. Figure 12 shows a diagram; on the left side, there are the characteristics of a good support material, and on the right side are the NPs options. Generally, the fabrication of an electrocatalyst requires a self-fabricated electrocatalyst, which usually involves the use of metalorganic frameworks (MOFs) [150], transition-metal compounds, carbon nanomaterials, foams, sponges, and aerogels.

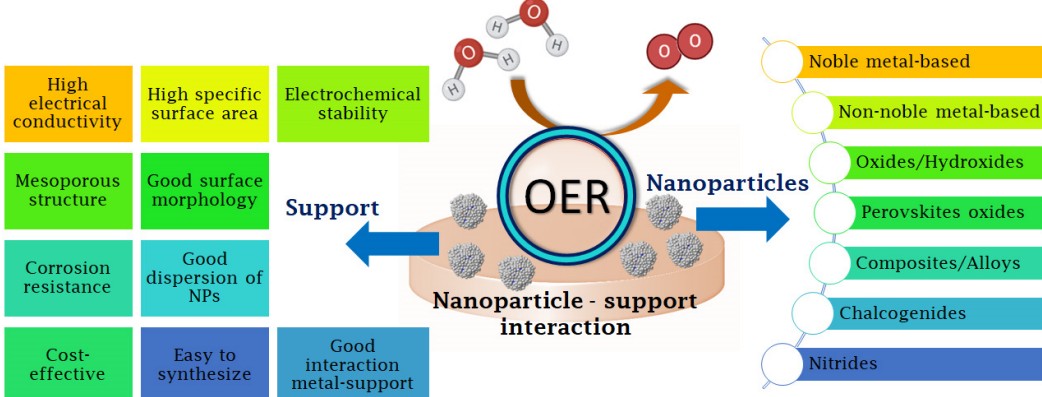

**Figure 12.** Schematic diagram of the characteristics of a good support material and diverse materials for nanoparticle synthesis.

Metal oxides are excellent materials in various electrochemical reactions due to their available active areas that increase the mobility of electrons in their porous metal networks. These materials have been studied for their low cost, abundance, distinctive structural properties, selectivity for electrochemical reactions over expensive metals, and efficiency. However, there are other challenges to be solved; one of them is related to their disordered growth and agglomeration, which reduce the electrocatalytic performance. Therefore, effective methods such as the use of carbon materials are required to control these difficulties. Complex oxides, including pyrochlore oxides, were recently been as support materials in alkaline OER applications. It is proposed that the adopted support helps to eject the generated electrons and reduce the energy barrier to intermediate formation. However, it remains unresolved whether similar approaches will be successful with an acidic OER; only a few stable and conductive oxides have been used, including Sn-, In-, W-, and Ti-based ones. For example, under the strong interaction between the Co ion cluster and the $TiO_2$ carrier, a charge transfer occurs between the Co site and the adjacent Ti atom, which initiates an efficient cocatalytic Co-Ti center, in which OH* and O* adsorb on the bridging site of Co and Ti, providing a low-energy barrier for $O_2$ generation. Therefore, Co-$TiO_2$ shows excellent intrinsic activity and durability against the OER [151].

$IrO_2$ supported in $WO_3$ with different concentrations was studied: IW-75 (75% $IrO_2$), IW-50 (50% $IrO_2$), and pure $IrO_2$. The bulk activity of IW-75 was determined to be approximately 1.15 times higher than that of the pure $IrO_2$ catalyst. The Tafel slope of $IrO_2$ is 131 mV dec$^{-1}$, while 104, 186, and 440 mV dec$^{-1}$ are measured for IW-75, IW-50, and $WO_3$ catalysts, respectively. These values confirm that IW-75 has a lower overpotential, a high current density (30.6 mA at 1.8 V), and a higher kinetic activity (following order: IW-75 > IW-50 > $IrO_2$ > $WO_3$) in 0.5 M $H_2SO_4$. According to the results obtained, the mixed-metal oxides $IrO_2$ and $WO_3$ are suitable candidates for the OER [152].

Carbon-based materials are often used as supports because of their large surface area and high conductivity. Currently, there are several types of carbon supports including graphenes, fullerenes, carbon nanofibers, carbon black, carbon balls, carbon nanotubes, porous carbon, etc., Figure 13, that have demonstrated remarkable synergetic actions on OERs. In fact, carbon nanomaterials are mainly used as additives/supports to improve the electrical conductivity of electrocatalysts.

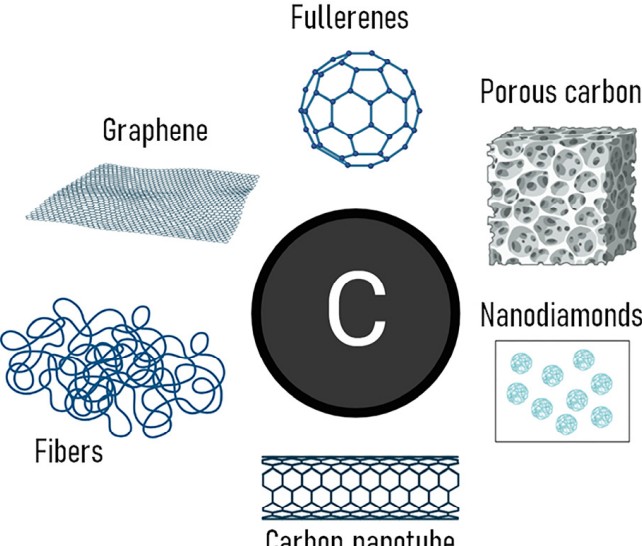

**Figure 13.** Different forms of carbon-based materials.

There are two popular procedures to produce carbon-supported NPs: (i) the carbon surface is modified using chemical functional groups and, subsequently, NPs are deposited on its surface. That is, the chemical bonds formed between the NPs and the supports would

also facilitate charge transfer at their interfaces [153]. (ii) The previously functionalized NPs are deposited on the surface of the carbon supports.

Commercial Pt-supported carbon (Vulcan XC 72-R) is commonly used as a cathodic catalyst for HER and ORR. The main problem is related to the poor durability of Pt/C in strong acid environment [154]. Carbon can oxidize to $CO_2$ under OER potentials and therefore is not suitable in practice as a support material in PEM electrolyzers. Consequently, other materials have recently been used to overcome these drawbacks between the catalyst-support interface.

The combination of metal oxides with different carbon supports to fabricate nanostructured materials is one of the systems implemented by the scientific community to study different electrode reactions [155]. Hazarika et al. obtained $Mn_2O_3$ NPs supported on carbon (Vulcan XC 72-R) for both the ORR and OER. The study performed in alkaline media showed that the ORR activity of $Mn_2O_3$/C was superior to that of Pt/C and Pd/C. The enhancement in the electrocatalytic activity of $Mn_2O_3$ was related to the influence of the carbon support interface [156]. $IrO_2$ was supported using multiwalled carbon nanotubes (MWCNTs). In addition, the electrochemical water oxidation activity of $IrO_2$@CNT series in 0.5 M $H_2SO_4$ electrolyte was evaluated. The overpotential at 10 mA cm$^{-2}$ of 0.5% $IrO_2$@CNT, 1.0% $IrO_2$@CNT, 1.5% $IrO_2$@CNT, and 2.0% $IrO_2$@CNT was 313, 306, 284, and 272 mV, respectively. The Tafel slope of 2.0% $IrO_2$@CNT (58 mV dec$^{-1}$) was lower than those of 0.5% $IrO_2$@CNT (89 mV dec$^{-1}$), 1.0% $IrO_2$@CNT (79 mV dec$^{-1}$), and 1.5% $IrO_2$@CNT (64 mV dec$^{-1}$). The OER stability of 2.0% $IrO_2$@CNT in 0.5 M $H_2SO_4$ showed that the overpotential only increased by 40 mV, showing good stability in acidic media [157].

Ultrafine metallic Co nanodots were embedded into the wall of N-doped carbon nanotubes (N-CNTs) grafted onto hexagonal vanadium nitride (VN). This hybrid structure (Co/N-CNT/VN) showed great properties such as a low overpotential of 240 mV at 10 mA cm$^{-2}$ and good electrochemical durability. N-CNT and VN with a robust structure provide ultrafast electron/ion transfer kinetics. The work showed a novel and simple approach that can be extended to the preparation of CNTs with other nanoparticles [158].

Boron and nitrogen codoped reduced graphene oxide (BN-rGO) was used as support for $IrO_2$ NPs in the OER [159]. An overpotential of 300 mV at 10 mA cm$^{-2}$, a high current density of 55 mA cm$^{-2}$ at 1.65 V, and a lifetime of 45 h in acidic medium (0.5 M $H_2SO_4$ solution) were found. The codoped BN showed an improvement in the current density obtained at 1.65 V, which was found to be twice that observed in the single-doped support (B-rGO) and ~2.5 times higher than that of $IrO_2$ in an undoped support. In the case of $IrO_2$-rGO, they show poor activity despite the high surface area of nanoparticles. While the catalytic activity of $IrO_2$-rGO can be attributed to the presence of the heteroatoms, it can lead to two phenomena in the lattice: (a) the opening of the band gap (electric field effect), and (b) strong metal–substrate interaction. When using conductive supports, the deposited metal centers influence their morphology, reactivity, and stability.

The structure and electronic properties of the support significantly influence the catalytic performance of the nanoparticles. The interfaces formed between NPs and supports have different coordination environments and thus serve as potential active sites for catalytic reactions. The electronic structure of the NPs, such as the *d*-band center, is affected by the supports. The choice of a suitable support to rationally control the position of the *d*-band center constitutes as a new avenue to regulate product selectivity.

## 5. Summary and Outlook

This paper gives an analysis of the fundamentals and strategies for the design of advanced electrocatalysts, essentially for the evolution of oxygen, where the reaction mechanisms and descriptors of this essential electrochemical process in an electrolyzer are discussed. The idea was to give a tour d'horizon of electrocatalytic materials whose composition starts from a single-metal center, so-called elemental, towards a more complex or multielemental composition, represented by materials, e.g., high-entropy alloys

(HEAs). In summary, the data generated in several significant contributions on acidic OER electrocatalyst compositions over the last decades are summarized in a table, including the design strategy and synthesis method used. The inclusion of the effect of support materials as a key complement in the design of electrocatalysts is also briefly considered. In this contribution, it can be seen that in recent years, significant progress has been made in understanding the mechanism of the OER. Nowadays, we have efficient tools, diverse strategies, and a wide panorama of approaches to implement and innovate the rational design of electrocatalysts. However, several challenges still need to be urgently addressed, especially in acidic media, before achieving a green transition. Some considerations and possible directions for further progress in this field are as follows:

1. The need not to lose sight of the objective of having a unifying descriptor of catalytic activity. However, it is equally essential to carry out theoretical and experimental investigations to understand the factors involved in poor stability in strongly acidic and corrosive environments. It is critically necessary to establish structure–stability relationships, so proposing descriptors that relate catalyst structure, activity, and stability is the key to rational design and will represent a valuable advance in the fundamentals of the OER.

2. Future research should try not only to break the scaling relationship but also to optimize the binding energies to achieve a lower overpotential. However, understanding and controlling the competition between different reaction mechanisms remains a major challenge.

3. Promote the development of electrocatalysts based on earth-abundant materials, such as low-cost, environmentally friendly transition metals with significant activity, to replace the use of very expensive and scarce iridium in acidic OER electrocatalysts, which is a bottleneck at the high-scale PEM-WE level.

4. In the meantime, electrocatalytic processes that are dynamic, surface-active sites, or active species can be generated during the electrochemical process, so it is necessary to pay special attention to the synthesis of the precatalytic material as well as to the operating conditions for its activation. Also, the self-reconstruction of the surface during the OER process can be beneficial or detrimental to the activity and/or stability of the electrocatalyst, usually by agglomeration, dissolution, and detachment of the catalyst. However, the structural reconstruction mechanisms are not entirely clear, so it is necessary to encourage the application of in situ and/or in operando techniques under electrochemical operating conditions, as these are a cornerstone in the development of highly active and stable electrocatalysts that allow a better understanding of the structural evolution of the materials in real time, as well as the nature of the key intermediates and their behavior. Therefore, these techniques allow in-depth understanding of the structure–activity–stability relationship of the OER process.

5. The recent perspective based on the concept of high entropy to design advanced materials is opening a new space of many possible combinations, leading us to rethink and explore new phenomena, theories, and applications. Certainly, this poses a great challenge, both theoretically and experimentally. Compared to minor component alloys, HEAs are still at a nascent stage. However, HEAs are promising candidates for improving the performance of both AWEs and PEM-WEs. This class of materials is of particular interest for replacing noble metals such as Ir, Ru, and Pt in acidic media. The synergistic effect of HEAs is mainly based on their elemental diversity; this requires new research methods, e.g., statistical methods, machine learning, data-driven combinatorial synthesis, high throughput, and screening techniques, for the efficient exploration of a suitable composition of HEAs as a cost-effective material and, in the not-too-distant future, more attractive properties might be discovered. However, there are still many fundamental concepts to be clarified and many questions and challenges to be addressed in the field of synthesis, characterization, and analysis of electrochemical data, identification of active sites, origin of high activity, etc.



6.   To improve data quality, standardized measurements of HER and OER properties are necessary as demonstrated by Hung et al. [160]. They recognized that 3D electrodes, such as metal foam materials (Ni, Fe, Co, IrNi-FeNi$_3$ on Ni, and other foams) are an interesting alternative, which exhibit high catalytic activities; however, the impact of parameters such as active area, thickness, porosity, capillarity, and purity need to be better understood. Some of them are not easy to determine experimentally. This contribution presents alternatives for ECSA calculation and highlights that an objective evaluation improves the reliability of the results enriches our fundamental understanding of electrocatalytic activity. Therefore, the implementation of standardized electrochemical parameters to evaluate the performance of electrocatalysts is an area of opportunity to ensure their industrial application.

Finally, we highlight the use of transition metals, which have shown great potential as OER electrocatalysts and their integration into emerging complex structures, e.g., HEAs. However, we recognize that there is superior progress in electrocatalysts in alkaline environments, so it is necessary to integrate great efforts to address the challenges of acidic conditions. We hope that this review can inspire and encourage research on advanced electrocatalysts, including high catalytic activity, long-term durability, and cost-effectiveness for OER in acidic conditions. A collective advance between the OER fundamentals, synthesis routes, characterization techniques, and electrochemical analysis will allow for the exploration of new functionalities for single to complex structures, which wait to be discovered for the development and commercialization of the PEM-WE, which has a great future in green hydrogen production as a clean and sustainable energy carrier.

Table 2. Review of OER electrocatalysts from single- to multielement materials in acid media.

| # | Catalysts | Element(s) | Support Material | Strategy Design | Synthesis Method | Overpotential @ 10 mA cm$^{-2}$ mV vs. RHE | Tafel Slope mV dec$^{-1}$ | Stability | Acidic Media | Ref. |
|---|---|---|---|---|---|---|---|---|---|---|
| **Single element** | | | | | | | | | | |
| 1 | Fe$_2$O$_3$ | Fe | Ti foil | Polymorphism structures | Spray pyrolysis | 650 | 56 | 24 h @ 10 mA cm$^{-2}$ | 0.5 M H$_2$SO$_4$ | [2] |
| 2 | Co$_3$O$_4$ crystalline | Co | Fluorine-doped tin oxide (FTO) | Engineering interface | Thermal annealing | 570 | 80 | 12 h @ 10 mA cm$^{-2}$ | 0.5 M H$_2$SO$_4$ | [161] |
| 3 | γ-MnO$_2$ | Mn | FTO | | Thermal decomposition | 489 | 79 | 8000 h @ 10 mA cm$^{-2}$ | 1.0 M H$_2$SO$_4$ | [112] |
| 4 | Co$_3$O$_4$ nanosheets | Co | Carbon paper | | Electroplating and calcination | 370 | 82 | 86.8 h @ 100 mA cm$^{-2}$ | 0.5 M H$_2$SO$_4$ | [162] |
| 5 | IrO$_x$@IrO$_2$ | Ir | *w/o* S | Structural engineering | Adams fusion | 309 | 45 | 6 h @ 10 mA cm$^{-2}$ | 0.1 M HClO$_4$ | [163] |
| 6 | Mesoporous iridium nanosheets | Ir | Vulcan XC-72 carbon | Mesoporous chemistry | Wet chemical/ micelles | 240 | 49 | 8 h @ 10 mA cm$^{-2}$ | 0.5 M H$_2$SO$_4$ | [164] |
| 7 | Mn$_3$O$_4$ nanoplates | Mn | --- | Crystal and geometric structure transformation | Rapid thermal annealing | 210 | 54.24 | 20 h @ 10 mA cm$^{-2}$ | 0.5 M H$_2$SO$_4$ | [113] |
| **Binary elements** | | | | | | | | | | |
| 8 | Fe$_5$Si$_3$ | Fe, Si | *w/o* S | Multiphase structure | Sintering process | 735 | 381.8 | 1000 cycles | 0.5 M H$_2$SO$_4$ | [165] |
| 9 | Ag-Co$_3$O$_4$ mesoporous | Co, Ag | FTO | Doping with foreign element (Ag) | Electrodeposition, hydrothermal, and calcination | 680 | 219 | 10 h @ 1.6 V Ag/AgCl | 0.5 M H$_2$SO$_4$ | [166] |
| 10 | C$_3$N$_4$-CNT-CF | C, N | Carbon fiber | Metal-free based | Annealing treatment | 570 | 129 | 14 h @ 1.63 V RHE | 0.5 M H$_2$SO$_4$ | [167] |
| 11 | TiO$_2$-modified MnO$_2$ | Mn, Ti | Polycrystalline gold disks | Engineering surface | Sputtering deposition | 510 @ 1 mA cm$^{-2}$ | 170 | 265 h @ 1.8 V RHE | 0.05 M H$_2$SO$_4$ | [168] |

Table 2. *Cont.*

| # | Catalysts | Element(s) | Support Material | Strategy Design | Synthesis Method | Overpotential @ 10 mA cm$^{-2}$ | Tafel Slope | Stability | Acidic Media | Ref. |
|---|-----------|-----------|-----------------|-----------------|------------------|---------------------------------|-------------|-----------|--------------|------|
| | | | | | | mV vs. RHE | mV dec$^{-1}$ | | | |
| **Binary elements** | | | | | | | | | | |
| 12 | Ag-Co$_3$O$_4$ (400) | Co, Ag | *w/o* S | Doping with foreign element (Ag) | Hydrothermal and annealed (400 °C) | 470 | 92 | 1000 cycles | 0.5 M H$_2$SO$_4$ | [169] |
| 13 | Co$_3$O$_4$/CeO$_2$ | Co, Ce | FTO | Engineering interface | Electrodeposition | 423 | 88.1 | - | 0.05 M H$_2$SO$_4$ | [170] |
| 14 | 2D MoS$_2$ nanosheets | Mo, S | *w/o* S | Polymorphism/ heteroatoms | Exfoliation and deposition | 420 | 322 | 2 h @ 10 mA cm$^{-2}$ | 0.5 M H$_2$SO$_4$ | [171] |
| 15 | W$_{0.57}$Ir$_{0.43}$O$_{3-\delta}$ | Ir, W | FTO | Structural engineering | Plasma synthesis | 370 | 125 | 0.56 h | 1.0 M H$_2$SO$_4$ | [172] |
| 16 | Cu$_{0.3}$Ir$_{0.7}$O$_\delta$ | Ir, Cu | Ti plate | Doping with foreign element (Cu) | Hydrothermal and annealing | 351 | 63 | 1.67 h @ 1.68 V RHE | 0.1 M HClO$_4$ | [173] |
| 17 | IrTe NTs | Ir, Te | Vulcan XC-72 carbon | Surface engineering | Galvanic replacement | 290 | 60.3 | 2000 cycles | 0.1 M HClO$_4$ | [174] |
| 18 | IrCu$_{0.77}$ | Ir, Cu | Vulcan XC-72 carbon | | Polyol method | 282 | 78.6 | --- | 0.1 M HClO$_4$ | [175] |
| 19 | RuNiOx | Ru, Ni | Carbon fiber paper | Dealloying treatment | Dip coating | 280 @ 50 mA cm$^{-2}$ | 51.91 | 10 h @ 1.5 V RHE | 0.5 M H$_2$SO$_4$ | [176] |
| 20 | IrMoOx nanofibers | Ir, Mo | --- | Electronic modulation | Electrospinning and calcination | 267 | 46.09 | 30 h | 0.5 M H$_2$SO$_4$ | [177] |
| 21 | Fe$_2$O$_3$/TiO$_2$ NWs/Ti | Fe, Ti | Ti foam | Structural engineering | Facile ion exchange process and calcination | 230 @ 1 mA cm$^{-2}$ | 110.7 | 20 h @ 1.9 V SCE | 0.5 M H$_2$SO$_4$ | [178] |
| 22 | Mn$_{0.73}$Ru$_{0.27}$O$_{2-\delta}$ | Ru, Mn | *w/o* S | Oxygen vacancies | Pyrolysis | 208 | 65.3 | 10 h @ 10 mA cm$^{-2}$ | 0.5 M H$_2$SO$_4$ | [179] |
| 23 | Zn-doped RuO$_2$ hollow nanorod | Ru, Zn | *w/o* S | Doping with foreign element (Zn) | Annealing process under air | 206 | 45.65 | 30 h @ 10 mA cm$^{-2}$ | 0.5 M H$_2$SO$_4$ | [180] |
| 24 | Mn-doped RuO$_2$ nanocrystals | Ru, Mn | *w/o* S | Doping with foreign element (Mn) | Annealing process under air | 158 | 42.94 | 10 h @ 10 mA cm$^{-2}$ | 0.5 M H$_2$SO$_4$ | [181] |
| 25 | Ru/MnO$_2$ | Ru, Mn | *w/o* S | Surface engineering/self-reconstruction | One-step cation exchange method | 161 | 29.4 | 10 h @ 10 mA cm$^{-2}$ | 0.1 M HClO$_4$ | [50] |

Table 2. *Cont.*

| # | Catalysts | Element(s) | Support Material | Strategy Design | Synthesis Method | Overpotential @ 10 mA cm$^{-2}$ | Tafel Slope | Stability | Acidic Media | Ref. |
|---|---|---|---|---|---|---|---|---|---|---|
| | | | | | | mV vs. RHE | mV dec$^{-1}$ | | | |
| Ternary elements | | | | | | | | | | |
| 26 | $Ni_{0.5}Mn_{0.5}Sb_{1.7}O_Y$ | Ni, Mn, Sb | Antimony-doped tin oxide | Phase restructuring | Sputter deposition | 672 | 85 | 168 h @ 10 mA cm$^{-2}$ | 1.0 M $H_2SO_4$ | [182] |
| 27 | $Ni_{40}Fe_{40}P_{20}$ | Ni, Fe, P | w/o S | Heteroatoms | Melting spinning | 540 | 40 | 30 h @ 10 mA cm$^{-2}$ | 0.05 M $H_2SO_4$ | [183] |
| 28 | $Mo-Co_9S_8$ | Mo, Co, S | Carbon cloth | Heteroatoms | solvothermal method | 370 | 90.3 | 24 h @ 1.6 V RHE | 0.5 M $H_2SO_4$ | [184] |
| 29 | F-doped $Cu_{1.5}Mn_{1.5}O_4$ nanoparticles | Cu, Mn, F | Porous Ti foil | Doping with foreign element (F) | Chemical synthesis and heat treatment | 320 @ 9.15 mA cm$^{-2}$ | 60 | 24 h @ 1.55 V RHE | 0.5 M $H_2SO_4$ | [185] |
| 30 | CP@NCNT | Co, P, N | N-doped CNT | Heteroatoms | Spray drying | 317 | 75 | 24 h @ 15 mA cm$^{-2}$ | 0.5 M $H_2SO_4$ | [186] |
| 31 | IrNiCu DNF/C | Ir, Ni, Cu | Vulcan XC-72 carbon | Nanoframe structure | Chemical etching | 303 | 48 | 2500 cycles | 0.1 M $HClO_4$ | [187] |
| 32 | $FeN_4$/NF/EG | Fe, N, C | Exfoliated graphene | Dual element-doping (N, C) | Electrodeposition and carbonization | 294 | 129 | 24 h @ 20 mA cm$^{-2}$ | 0.5 M $H_2SO_4$ | [188] |
| 33 | Co-doped IrCu | Ir, Co, Cu | Vulcan XC-72 carbon | Doping with foreign element (Co) | Chemical etching | 293 | 50 | 2000 cycles | 0.1 M $HClO_4$ | [189] |
| 34 | W-Ir-B | Ir, W, B | w/o S | Biphasic structure | Arc melting and drop casting | 291 | 78 | 120 h @ 100 mA cm$^{-2}$ | 0.5 M $H_2SO_4$ | [190] |
| 35 | $RuO_2/(CoMn)_3O_4$ | Ru, Co, Mn | Carbon cloth | Interface engineering | Hydrothermal process | 270 | 77 | 24 h @ 10 mA cm$^{-2}$ | 0.5 M $H_2SO_4$ | [116] |
| 36 | HNC-Co | Co, N, C | Carbon paper | Dual element-doping (N, C) | Polymerization, reduction, and pyrolysis | 265 | 85 | 100 h | 0.5 M $H_2SO_4$ | [191] |
| 37 | $SrTi_{0.67}Ir_{0.33}O_3$ | Ir, Sr, Ti | w/o S | Ir doping | Polymerized complex method | 247 | 43 | 20 h @ 10 mA cm$^{-2}$ | 0.1 $HClO_4$ | [192] |
| 38 | $C-RuO_2-RuSe-5$ | Ru, Se, C | w/o S | Inducing interstitial atoms | Hydrothermal method | 212 | 49.5 | 30 h @ 10 mA cm$^{-2}$ | 0.5 M $H_2SO_4$ | [193] |
| 39 | Ru NCs/$Co_2P$ Hollow microspheres | Ru, Co, P | w/o S | Heteroatoms | Hydrothermal | 197 | 89 | 10 h | 0.5 M $H_2SO_4$ | [194] |

**Table 2.** *Cont.*

| # | Catalysts | Element(s) | Support Material | Strategy Design | Synthesis Method | Overpotential @ 10 mA cm$^{-2}$ | Tafel Slope | Stability | Acidic Media | Ref. |
|---|---|---|---|---|---|---|---|---|---|---|
| | | | | | | mV vs. RHE | mV dec$^{-1}$ | | | |
| Quaternary elements | | | | | | | | | | |
| 40 | InFeCo-CCP | Fe, Co, In, N | *w/o* S | Organic/inorganic polymeric | Coordination-substitution polymerization | 710 | 99 | 48 h @ 1.75 V RHE | 0.5 M H$_2$SO$_4$ | [195] |
| 41 | P-NSC/Ni$_4$Fe$_5$S$_8$ | Ni, Fe, N, S | *w/o* S | Heteroatoms/porous | Pyrolysis (1000 °C) | 550 | 72.1 | 10,000 cycles | 0.5 M H$_2$SO$_4$ | [196] |
| 42 | Fe$_{35}$Ni$_{35}$Co$_{10}$P$_{20}$ | Fe, Ni, Co, P | *w/o* S | Amorphous multialloy | Arc-melting technique | 497 | 79 | 20 h @ 1.73 V RHE | 0.5 M H$_2$SO$_4$ | [197] |
| 43 | Ni$_{42}$Li$_2$O$_5$ | Steel (Ni, Fe, Mn), Li | AISI Ni42 steel | Surface engineering | Electrooxidation | 445 | 260 | 5.6 h @ 10 mA cm$^{-2}$ | 0.05 M H$_2$SO$_4$ | [198] |
| 44 | Mn-doped FeP/Co$_3$(PO$_4$)$_2$ | Fe, Co, Mn, P | Carbon cloth | Heteroatoms | Hydrothermal | 390 | 472 | 10,000 cycles | 0.5 M H$_2$SO$_4$ | [199] |
| 45 | Ba[Co-POM]/CP | Co, W, Ba, P | Carbon paste | Polyoxometalate | Metathesis | 361 | 97 | 24 h @ 1.48 V RHE | 1.0 M H$_2$SO$_4$ | [200] |
| 46 | CoMoNiS-NF-31 | Co, Mo, Ni, S | Nickel foam | Heteroatoms/hierarchical structure | One-pot hydrothermal | 228 | 78 | --- | 0.5 M H$_2$SO$_4$ | [121] |
| Multi-elements | | | | | | | | | | |
| 47 | FeCoNiMnW | Fe, Co, Ni, Mn, W | Carbon paper | Multimetal alloy | Electrodeposition | 332 | 145 | 1 h | 0.5 H$_2$SO$_4$ | [201] |
| 48 | Al$_{89}$Ag$_1$Au$_1$Co$_1$Cu$_1$Fe$_1$Ir$_1$Ni$_1$Pd$_1$Pt$_1$Rh$_1$Ru$_1$ | Ir, Ru, Pt, Pd, Au, Rh, Ag, Al, Co, Cu, Fe, Ni, | *w/o* S | Multimetal alloy | Arc melting and one-step dealloying | 258 | 84.2 | 11.11 h @ 10 mA cm$^{-2}$ | 0.5 H$_2$SO$_4$ | [202] |
| 49 | FeCoNiIrRu | Ir, Ru, Fe, Co, Ni | Carbon nanofibers | Multimetal alloy | Electrospinning | 241 | 153 | 4 h | 0.5 H$_2$SO$_4$ | [203] |
| 50 | Al$_{96}$Ni$_1$Co$_1$Ir$_1$Mo$_1$ | Ir, Al, Ni, Co, Mo | Carbon powder | Multimetal alloy | Induction-melting furnace and chemical etching | 233 | 55.2 | 7000 cycles | 0.5 H$_2$SO$_4$ | [126] |
| 51 | IrPdRhMoW | Ir, Pd, Rh, Mo, W | *w/o* S | Multimetal alloy/structural engineering | Oil-phase method | 188 | --- | 100 h @ 100 mA cm$^{-2}$ | 0.5 H$_2$SO$_4$ | [127] |

*w/o* S: without support.

**Author Contributions:** Writing—original draft preparation and conceptualization: L.C.O., A.H., R.B. and N.A.-V.; writing—review and editing: L.C.O., A.H., D.L.H.-A., R.B. and N.A.-V. All authors have read and agreed to the published version of the manuscript.

**Funding:** Not applicable.

**Data Availability Statement:** Not applicable.

**Acknowledgments:** A. Higareda acknowledges CONAHCYT for the postdoctoral research fellowship 2022(1), application number 2415291. Diana L. Hernández-Arellano acknowledges CONAHCYT for the doctoral research fellowship, number 814166. N. Alonso-Vante acknowledges financial support from the European Union (ERDF) 'Région Nouvelle Aquitaine'.

**Conflicts of Interest:** The authors declare no conflict of interest.

**Abbreviations**

The following abbreviations are used in this manuscript:

| | |
|---|---|
| AEM | Alkaline electrolyte membrane |
| AMO | Amorphous manganese oxides |
| AWE | Alkaline water electrolyzers |
| BEDPs | Binding energy distribution patterns |
| CPET | Concerted proton–electron transfer |
| CUS | Coordinatively unsaturated |
| DEMS | Differential electrochemical mass spectrometry |
| DFT | Density functional theory |
| ESSI | Electrochemical step symmetry index |
| HCP | Hexagonal close-packed structure |
| HEAs | High-entropy alloys |
| HER | Hydrogen evolution reaction |
| IMCs | Intermetallic compounds |
| LDH | Double-layered hydroxides |
| LOM | Lattice oxygen mechanism |
| MOFs | Metalorganic frameworks |
| MD | Molecular dynamics |
| MWCNTs | Multiwalled carbon nanotubes |
| NAP | Near ambient pressure |
| NPs | Nanoparticles |
| OER | Oxygen evolution reaction |
| ONB | Oxygen nonbonding states |
| PEM | Proton exchange membrane |
| PEM-WE | Proton exchange membrane water electrolyzer |
| RDS | Rate-determining step |
| RHE | Reversible hydrogen electrode |
| RPS | Rate-determining potential step |
| SHE | Standard hydrogen electrode |
| SI-SECM | Surface interrogation scanning electrochemical microscopy |
| STEM-EDS | Scanning transmission electron microscopy–energy dispersive X-ray spectroscopy |
| UV-vis | Ultraviolet–visible |
| XAS | X-ray absorption spectroscopy |
| XPS | X-ray photoelectron spectroscopy |
| XRD | X-ray diffraction |

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
