# Peer review of "Advanced Electrocatalysts for the Oxygen Evolution Reaction: From Single- to Multielement Materials"

_catalysts, doi:10.3390/catal13101346_

Round 1
Reviewer 1 Report
See attached file.

See attached file.
Author Response
See attachement.
Reviewer 2 Report
In this work, the recent progress in the fundamentals and strategies for the design of advanced electrocatalysts for oxygen evolution, reaction mechanisms, and OER descriptors. The discussion on the elemental composition of the OER electrocatalysts from single to multi-elemental, as well as high entropy alloys. The effect of support materials on the electronic properties of the catalysts is also presented in this review. A good effort has been made to provide an overview of the current progress. Therefore, I recommend its publication after major revision.
The following comments are provided to further improve the quality of the manuscript:
1. In the second part “2. Fundamentals of advanced electrocatalysts design for oxygen evolution reaction”, the authors present two mechanisms of the OER, including AEM and LOM. However, another mechanism named “oxide path mechanism (OPM) [Nature Catalysis, 2021, 4, 1012]” is ignored by the authors, and should be added to the manuscript.
2. In the Abstract, the authors stated that the design of acid-stable electrocatalysts with low overpotential and excellent stability for OER constitutes an important activity in electrocatalysis. However, the summary in part “3. Oxygen evolution reaction electrocatalysts ” are mostly transition-metal based-electrocatalysts in alkaline media. More progress on the electrocatalysts in acid media should be added to the manuscript.
3. This reviewer suggests the authors add some figures into part “3. Oxygen evolution reaction electrocatalysts ” from the representative works.
4. The relationships between single-component, binary-component, ternary-component, quaternary-component, and multi-component should be clarified (the advantages and disadvantages).
5. In part “4. Effect of supports”, this reviewer suggests the authors summarize the support materials from the perspective of their functions. For example, the interaction between the electrocatalyst and support improves the intrinsic activity, increases the number of active sites, and promotes the stability of the electrocatalyst.
Author Response
See attachement

Reviewer 3 Report
Reviewer #1: In this manuscript, the authors systematically reviewed Advanced Electrocatalysts for the Oxygen Evolution Reaction: From Single to Multi-Element Materials. The entire article is well comprehensive with consulting substantial references. It is worth mentioning that the author identified the critical limitations for the present singles catalysts and provided prospective outlooks. However, the insufficient descriptions of these limit the readers' interest and further exploration. There are also some other minor issues as following mentioned. Overall, the manuscript is recommended for publication in Catalysts after minor revisions.
1. Please check on Table 1, Equation should be more visible for authors. For example line 216-224.
2. There is only one figure in Figure 1, impairing the readability. I would suggest splitting them to make the relative content close to the depicted figure.
3. The authors should give some examples to support this paper such as Nano Energy, vol. 73, pp. 104788-1 – 104788-8 (2020)
4. Graphic abstract (or commonly known as TOC graph) should have adequate resolution and clarity; Confirm that all text is legible at the final size.
5. More electrochemical characterizations descriptions are necessary to enable readers to better understand the performance of the catalysts.
6. Please improve the Figure 8 quality, tailoring the inappropriate or unnecessary parts.
7. The author mentioned the advanced operando characterizations techniques used to probe the real catalytic mechanism behind it. These recently published relevant references should be cited (Trends in Chemistry, vol. 4, no. 12, pp. 1065 – 1077 (2022).;
Other literature
Enhanced oxygen reduction activity and stability of double-layer nitrogen-doped carbon catalyst with abundant Fe-Co dual-atom sites
A high-entropy phosphate catalyst for oxygen evolution reaction
Electrocatalysts for acidic oxygen evolution reaction: Achievements and perspectives
Disclosure of Charge Storage Mechanisms in Molybdenum Oxide Nanobelts with Enhanced Supercapacitive Performance Induced by Oxygen Deficiency
good
Author Response
See attachement.
